# One-step Diffusion Solver for Non-binary Integer Linear Programming

## Abstract

Integer linear programming, a fundamental NP-hard problem with broad applications in science and engineering, has gained growing attention in the machine learning community. Yet, progress on effective end-to-end solvers remains limited, largely due to difficulties in enforcing constraints and integrality. Most existing work focuses on binary integer linear programming problems, while generalizing to bounded, non-binary cases often requires transformations that significantly increase problem size and computational costs. Even for purely binary problems, inference time is often prohibitively long, restricting applicability to real-world scenarios. To tackle the aforementioned problems, we propose three one-step diffusion-based approaches, i.e., CMILP, SCMILP and MFILP, inspired by the popular consistency, shortcut and meanflow training techniques. Our methods can further handle non-binary integer problems using a newly proposed iterative integer projection (IIP) layer, eliminating the need for the costly problem transformation. To further improve the solution quality, an objective-guided sampling with momentum scheme is proposed. Experiments demonstrate that our approach outperforms existing learning-based methods on both binary and non-binary instances and shows strong scalability compared to traditional solvers. Source code and detailed protocols will be made publicly available.

## 1 Introduction

Integer Programming (IP) (Schrijver, 1998) is a class of optimization problems in the field of operations research, where some or all of the decision variables are constrained to be integers. These problems play a crucial role in various domains, such as production planning (Pochet & Wolsey, 2006), resource allocation (Zoltners & Sinha, 1980), and scheduling (Ryan & Foster, 1981). However, as an NP-hard problem, IP is generally very difficult to solve. In recent decades, researchers have primarily relied on heuristic methods such as branch-and-bound (Wolsey, 2020), cutting-plane methods (Ceria et al., 1998), and large neighborhood search algorithms (Ahuja et al., 2002) to address this challenge. These methods are typically computationally expensive, especially for large-scale problems, where the search space grows exponentially, significantly increasing the difficulty of solving the problem.

With the success of machine learning, recent studies have begun focusing on solving IP problems using data-driven approaches (Gasse et al., 2019), where neural networks are used to predict solutions that both minimize the objective function and ensure feasibility of the solutions. To address this issue, Wang et al. (2022) proposed a differentiable IP solver that uses Gumbel-Softmax to ensure that integer constraints do not interfere with the learning of the optimal objective. Meanwhile, Zeng et al. (2024) leveraged deep diffusion models to nearly perfectly satisfy 0-1 integer constraints. However, there are still several notable issues: 1) Although Zeng et al. (2024) excels at generating feasible solutions, the inference speed of diffusion models is very slow, leading to a loss of the efficiency advantages deep learning should offer compared to traditional solvers like Gurobi and COPT; 2) Most existing IP neural solvers are limited to 0-1 integer programming and fail to extend neural IP solvers to more general integer constraints. Though bounded IP problems can be transformed to binary integer programming problems, the problem scale grows exponentially with the variable bounds and will bring about large computational burdens.

To tackle these problems, in this paper, we propose three one-step diffusion-based integer linear programming solvers. The structure of the solvers is visualized in Fig. 1. One-step solvers can

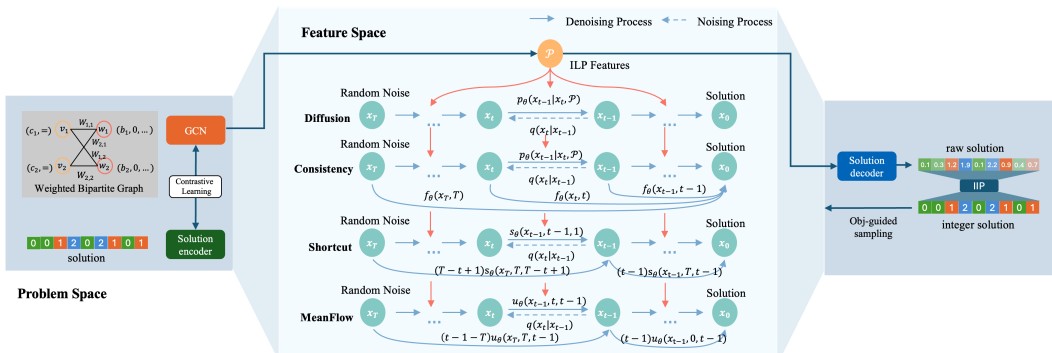

Figure 1: Illustration of the proposed one-step diffusion solvers for non-binary ILP.

finish solving far faster compared to traditional solvers and vanilla-diffusion-based solvers with comparable performance. To enhance the solution's feasibility, we adopt the objective-guided sampling methods. The momentum mechanism is further introduced to boost the effectiveness of the objective-guided sampling. We evaluate our methods on not only classic binary integer linear programming problems, but also two types of non-binary integer linear programming problems. Experimental results demonstrate the superiority of our method over original diffusion-based methods. **In short, this work contributes in the following aspects:**

1) Departure from previous works that employ a two-stage method to handle infeasible solutions (Nair et al., 2021) or use extensive diffusion iterations (Zeng et al., 2024) to obtain feasible solutions. In this paper, we propose three one-step diffusion-based solvers under an end-to-end paradigm for ILP problems, namely CMILP, SCMILP and MFILP. The proposed solvers achieve higher solution feasibility compared to previous neural solvers, reaching nearly 100% on binary ILP problems without resorting to traditional algorithms for post-processing.

2) For the first time, to our best knowledge, we extend the binary 0-1 ILP neural solver to the non-binary case for feasible solution prediction, in which we introduce a new Iterative Integer Projection (IIP) layer defined across the entire real domain, capable of approximating the real integer within a few iterations. We find that using a small number of projection iterations during training, and more iterations during testing, leads to better performance.

3) We propose and rethink the guidance in the diffusion model for ILP, as presented in Zeng et al. (2024), from the perspective of non-convex optimization. We show that previous guidance methods can be viewed as a special case of gradient descent (with only a single optimization step). Based on this insight, we introduce a sampling method based on gradient descent and a momentum-based gradient descent approach to improve the sampling process.

## 2 RELATED WORKS

**Diffusion-based Models** The diffusion model (Ho et al., 2020) is a popular generative model that has been actively applied to solve various optimization problems (Sun & Yang, 2023; Li et al., 2023). It uses a noising and denoising procedure to accurately capture the target distribution. To accelerate the inference speed of the diffusion model, the consistency model (Song et al., 2023) is devised by posing the consistency function onto the variable trajectory. Flow matching (Lipman et al., 2023) generalizes the diffusion model and generates in a continuous normalizing flow-based paradigm. The shortcut model (Frans et al., 2024) is a newly devised one-step diffusion model that takes the step size as the conditional input to permit large step sampling. Instead of focusing on instantaneous velocity as in flow matching models, meanflow (Geng et al., 2025a) tries to learn the average velocity.

**(Mixed) Integer Linear Programming and its Traditional Solvers** Mixed Integer Linear Programming (MILP) (Bénichou et al., 1971) is a fundamental optimization technique widely used across various fields, including operations research and supply chain management. Traditional MILP solvers include branch-and-bound (Wolsey, 2020), branch-and-cut (Mitchell, 2002), and cutting-plane (Ceria et al., 1998) methods. The branch-and-bound method systematically divides the solution space into

subproblems and eliminates infeasible solutions based on bounds, while branch-and-cut enhances this approach by incorporating cutting planes to improve computational efficiency. Cutting-plane methods iteratively refine the feasible region by adding linear inequalities, thus reducing the search space. Additionally, simplex-based methods have been adapted for MILP through algorithms such as the dual simplex method (Banciu, 2011). While these traditional solvers are effective, they often struggle with large-scale problems, where computational time grows exponentially. This challenge has spurred the development of hybrid approaches that combine traditional solvers with metaheuristics, constraint programming, and machine learning techniques to improve efficiency in solving complex, large-scale MILP problems. In contrast, Neural Solvers offer faster and more scalable solutions by learning from data, enabling quicker problem-solving without the need for manual adjustments.

**Neural Solver for IP**    (Mixed) Integer Linear Programming (ILP), as a widely used mathematical programming problem, has attracted a great deal of attention from the machine learning community (Zhang et al., 2023). One line of research tries to substitute ML models with key parts of traditional algorithms to improve solving efficiency. A significant portion of this work focuses on learning heuristic policies for tasks such as selecting variables to branch on (Scavuzzo et al., 2024), choosing cutting planes (Puigdemont et al., 2024), and more (Labassi et al., 2022). Another line of research leverages ML models to predict solutions and adopts traditional methods as post-processing techniques to retrieve feasible solutions. For example, Neural Diving (Nair et al., 2020) predicts a partial solution based on coverage rates and utilizes neural networks to determine which predicted variables to fix, while Han et al. (2023a); Ye et al. (2023) builds upon this work by adopting search methods to improve solution quality. Tang et al. (2025) deals with non-binary ILP by introducing an integer correction layer at the cost of extra parameters. Most of these works focus more on integer prediction and do not directly address the satisfaction of linear constraints. As a result, they are not end-to-end models and rely on heuristic search to satisfy the inequality constraints. In this paper, we attempt to propose an end-to-end model to get feasible solutions using merely machine learning techniques. Acceleration is expected due to the speed advantage neural networks usually bring about.

## 3 METHODOLOGY

### 3.1 REPRESENTATIONS OF ILP WITH PROJECTED GRAPH NEURAL NETWORKS

**ILP representation.** Integer Linear Programming (ILP) Problem is a type of optimization problem that seeks an integer-valued solution that minimizes a linear objective under linear constraints. All integer linear programming problems can be transformed to the following form:

$$\min_{\mathbf{x}} \mathbf{c}^\top \mathbf{x}, \quad \text{s.t.} \ \mathbf{A}\mathbf{x} \leq \mathbf{b}, \mathbf{x} \in \mathbb{Z}^n, \mathbf{b} \in \mathbb{R}^m, \mathbf{A} \in \mathbb{R}^{m \times n} \tag{1}$$

where there are $n$ variables and $m$ constraints. Given that SCIP (Bolusani et al., 2024) already provides a mature algorithm for this transformation, we only tackle such problems during model training. Following Gasse et al. (2019), we represent ILP as a weighted bipartite graph, where variable and constraint nodes form two disjoint sets and the bipartite graph weights encode the constraint matrix $\mathbf{A}$. This representation allows us to leverage a graph neural network for feature extraction. More specifically, in this paper, we adopt the network architecture implemented by Nair et al. (2021).

**Model architecture.** Considering the discrete nature of the solutions of ILP instances, a projection should be applied to map the variables to a continuous feature space, which is implemented via a transformer in our model. As proven in Nair et al. (2021), when extracting features from the problem instances and problem solutions, we should also ensure the alignment between them to enhance model performance. Motivated by CLIP (Radford et al., 2021), we adopt a contrastive learning approach to better match the continuous ILP problem features (node features of weighted bipartite graphs) and the solution features. The CLIP-style encoder is pretrained to extract robust instance features independently of solver training.

Secondly, a neural solver is applied to solve the instance in the feature space. We utilize generative-model-based solvers here to learn the solution distribution given problem instances. This type of solvers are proven effective on various combinatorial datasets (Li et al., 2024). The continuous nature of the feature space permits a smooth adoption from the well-developed image generative models. The specific diffusion solvers will be introduced in the following sections. The backbone of this solver is a transformer encoder that learns the solution distribution. The solution features generated

previously are treated as the targets, while the ILP instance features serve as the conditional inputs. The time $t$ of the diffusion trajectory is encoded using the sinusoidal embedding. Since our goal is to capture the underlying solution distribution, we construct the training set by collecting 500 optimal and sub-optimal solutions, allowing for a richer representation of the data distribution.

Finally, a solution decoder is applied to project the solution features back to the original solution space. The final solution is reconstructed by a combination of the predicted solution features and the ILP problem features. The solution decoder is trained jointly with the diffusion model.

The model is trained to minimize the reconstruction error, the diffusion loss and a feasibility penalty. The reconstruction error $\mathcal{L}_{\text{recon}}$ measures the model's ability to capture the mapping between the problem space and the feature space. For binary variables, the cross entropy loss is adopted for the reconstruction error. For non-binary cases, we choose the MSE loss to evaluate the reconstruction gap. The diffusion loss $\mathcal{L}_{\text{XXILP}}$ aims to enhance the learning of the solution distribution conditioned on the problem distribution. To further enforce constraint satisfaction, we introduce a feasibility penalty $\mathcal{L}_{\text{penalty}} = \frac{1}{m}\sum_{i=1}^{m}\max(a_i^T\hat{x} - b_i, 0)$, which specifically addresses linear constraint violations. Integrality is handled separately through the Iterative Integer Projection described below. Experimental results confirm that incorporating the feasibility penalty significantly improves constraint satisfaction in our solver. The final training loss is shown as follows:

$$\mathcal{L} = \mathcal{L}_{\text{recon}} + \mathcal{L}_{\text{XXILP}} + \lambda_{\text{penalty}}\mathcal{L}_{\text{penalty}}, \tag{2}$$

where $\lambda_{\text{penalty}}$ is the penalty coefficient.

**Iterative Integer Projection (IIP) for General ILP.** Most existing studies (Zeng et al., 2024; Li et al., 2023) focus on binary integer linear programming (BILP) problems (i.e. $\mathbf{x} \in \{0, 1\}^n$) due to their relative simplicity. While it is theoretically possible to transform any bounded integer linear programming instance into a binary form through binary encoding techniques (Nair et al., 2021), such transformations often lead to an exponential increase in problem size. This scaling significantly impacts computational efficiency, increasing both solving time and memory costs.

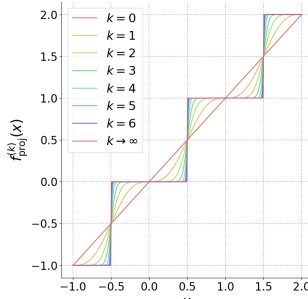

In this work, we turn our focus to non-binary ILP problems—a direction that has received comparatively little attention. Tackling such problems requires a differentiable mechanism for approximating non-binary integer variables. While the Sigmoid function is widely used for relaxing binary variables, extending this idea to the non-binary case calls for a new projection function that meets several criteria: it must be differentiable, defined over the real domain, and capable of rapidly converging to integer values within only a few iterations. Guided by these considerations, we introduce the following integer projection function:

Figure 2: Visualization of the Iterative Integer Projection $f_{\text{proj}}^{(k)}$. As the iteration $K$ increases, the projected results gradually converge to integers.

$$f_{\text{proj}}^{(0)}(\mathbf{x}) = \mathbf{x}, \text{ and } f_{\text{proj}}^{(k)}(\mathbf{x}) = f_{\text{proj}}(f_{\text{proj}}^{(k-1)}(\mathbf{x})) \,\forall k < K \quad \text{where } f_{\text{proj}}(\mathbf{x}) = \mathbf{x} - \frac{\sin(2\pi\mathbf{x})}{2\pi} \tag{3}$$

Here, $K \geq 0$ represents the number of projection iterations for this layer. Through this recursive iteration, we can approximate the integer solution of the output $\mathbf{x}$ in a differentiable manner. Fig. 2 demonstrates how the above function approximates the rounding function in finite iterations. We use this function to replace the Sigmoid function to approximate integer values throughout the real domain. The projection is applied once during training for training efficiency and applied multiple times during testing for approximation accuracy.

## 3.2 ONE STEP DIFFUSION MODELS FOR FOR INTEGER LINEAR PROGRAMMING

In this section, we start to devise diffusion-based solvers to address integer linear programming problems. Unlike purely supervised-learning-based solvers that aim to predict a single optimal solution, diffusion-based methods learn the distribution of feasible solutions $\mathbf{x}$ given instances $\mathcal{P}$, i.e., $q(\mathbf{x} \mid \mathcal{P})$. This distribution is modeled by transforming Gaussian noise through a learned generative process.

However, solvers based on the vanilla diffusion model suffer from a long inference time, even compared to traditional solvers (Zeng et al., 2024). This limits this model's practical value. Hence, we propose using the consistency model (Song et al., 2023), shortcut model (Frans et al., 2024) and mean flow model Geng et al. (2025a), the speed-up version of the diffusion model, to address ILP problems. The detailed introduction of shortcut and mean flow models are put in the appendix.

**CMILP**. Vanilla diffusion-based solver is comprised of a noising and denoising process. The noising process takes the initial solution $\mathbf{x}_0$ and progressively introduces noise to generate trajectory $\mathbf{x}_{1:T} = \mathbf{x}_1, \mathbf{x}_2, ..., \mathbf{x}_T$. Specifically, the noising process is modeled as $q(\mathbf{x}_{1:T}|\mathbf{x}_0) = \prod_{t=1}^{T} q(\mathbf{x}_t|\mathbf{x}_{t-1})$, where each transition is formulated as a Gaussian distribution, i.e.

$$q(\mathbf{x}_t|\mathbf{x}_{t-1}) = \mathcal{N}(\mathbf{x}_t; \sqrt{1-\beta_t}\mathbf{x}_{t-1}, \beta_t\mathbf{I}) \qquad (4)$$

where $\beta_t \in [0, 1]$. We further define $\alpha_t = 1 - \beta_t, \bar{\alpha}_t = \Pi_{i=1}^{t}\alpha_i$. Using reparametrization trick, we can sample $\mathbf{x}_t$ through $\mathbf{x}_t = \sqrt{\alpha_t}\mathbf{x}_{t-1} + \sqrt{1-\alpha_t}\epsilon_t$, where $\epsilon_t \sim \mathcal{N}(0, \mathbf{I})$. During testing, we recreate a true sample $\mathbf{x}_0$ from a Gaussian noise input $\mathbf{x}_T$ by reversing the above noising process. Ho et al. (2020) proves that the denoising process is modeled as:

$$\mathbf{x}_{t-1} = \frac{1}{\sqrt{\alpha_t}}(\mathbf{x}_t - \frac{1-\alpha_t}{\sqrt{1-\bar{\alpha}_t}}\epsilon_t) + \sqrt{\frac{1-\bar{\alpha}_{t-1}}{1-\bar{\alpha}_t}\beta_t}\mathbf{z}, \mathbf{z} \sim \mathcal{N}(0, \mathbf{I}) \qquad (5)$$

We then train a neural network to approximate this distribution. Consistency model further introduces the consistency function $f_\theta$ to formulate the trajectory. $f_\theta$ is characterized by: 1) boundary condition: $f_\theta(\mathbf{x}_\epsilon, \epsilon) = \mathbf{x}_\epsilon$, where $\epsilon$ is the initial timestep; 2) self-consistency properties: $f_\theta(\mathbf{x}_t, t) = f_\theta(\mathbf{x}_{t'}, t'), \forall t, t' \in [\epsilon, T]$. The consistency model requires all variables along the noising and denoising route to yield the same value for the consistency function. The introduction of the consistency function shortens the inference schedule to one or a few timesteps, greatly reducing the inference time. This makes the consistency model more practical in real-world settings.

Considering the characteristics of integer programming, we choose the mapping to the solution distribution as the consistency function. This consistency function follows both boundary conditions and self-consistency properties because the solution distribution is determined by the problem features. Since the solution $\mathbf{x}^*$ is explicit given the problem instance, we can integrate $\mathbf{x}^*$ into the loss for better training instead of focusing on the gap between $f_\theta$ of two diverse timesteps:

$$\mathcal{L}_{\text{CMILP}}^{N_t}(\theta) = \mathbb{E}\left[d(f_\theta(\mathbf{x}'_{t_n}, t_n, \mathcal{P}), \delta(\mathbf{x} - \mathbf{x}^*)) + d(f_\theta(\mathbf{x}_{t_{n+1}}, t_{n+1}, \mathcal{P}), \delta(\mathbf{x} - \mathbf{x}^*))\right] \qquad (6)$$

where $d(\cdot, \cdot)$ is a distance function, $N_t$ represents the time scheduler, $\delta(\cdot)$ is Dirac delta and $\mathcal{P}$ is the problem instance. $x_t$ and $x'_t$ are sampled from two independent and identically distributed trajectories, as in the original consistency loss. Its minimization is achieved only if consistency holds across all possible trajectories, yielding the optimal solution distribution.

### 3.3 OBJECTIVE GUIDED SAMPLING FOR DIFFUSION MODEL

Constraint satisfaction and objective minimization are two core problems in constrained optimization. We attempt to incorporate them into diffusion's sampling to further boost models' performance. We follow Graikos et al. (2023) to utilize the learned model $p(\mathbf{x}|\mathcal{P})$ as a sampling prior to achieve this. The conditional information $\mathbf{y}^*$ is incorporated using a constraint function $c(\mathbf{x}, \mathbf{y}^*)$ to regulate the posterior distribution. The target posterior $p_\theta(\mathbf{x}|\mathbf{y}^*)$ is hence modeled as $Zp_\theta(\mathbf{x}|\mathcal{P})c(\mathbf{x}, \mathbf{y}^*|\mathcal{P})$. We introduce an approximate variational posterior $q(\mathbf{x}|\mathcal{P})$ to estimate the target posterior. Following the derivation in Li et al. (2024), if we approximate $q(\mathbf{x}|\mathcal{P})$ as a point estimate $\delta(\mathbf{x} - \boldsymbol{\eta})$, where $\boldsymbol{\eta}$ point estimate's parameter, we can minimize the following property to learn the target posterior.

$$F = \mathbb{E}_{q(\mathbf{h}|\boldsymbol{\eta},\mathcal{P})}\left[\log\frac{q(\mathbf{h}|\boldsymbol{\eta},\mathcal{P})}{p_\theta(\boldsymbol{\eta},\mathbf{h}|\mathcal{P})} - l(\boldsymbol{\eta};\mathcal{P}) - \log Z - \mathbf{y}^*\right] \qquad (7)$$

where Z is a constant normalization factor, $\mathbf{h} = \mathbf{x}_1, \ldots, \mathbf{x}_T$ are the latent variables and $l(\cdot;\mathcal{P})$ is defined as follows:

$$\mathbf{y}^* = \min_{\mathbf{x}} l(\mathbf{x};\mathcal{P}) \quad \text{where} \quad l(\mathbf{x};\mathcal{P}) \triangleq \mathbf{c}^\top\mathbf{z} + \sum \max(a_k^\top\mathbf{z} - b_k, 0), \text{ and } \mathbf{z} = \text{Decoder}(\mathbf{x}) \qquad (8)$$

where $a_k^\top$ is the kth row in the constraint matrix $\mathbf{A}$ and $b_k$ is the kth value in the constraint vector $\mathbf{b}$. From the diffusion process, we have $q(\mathbf{h}|\mathbf{x},\mathcal{P}) = \prod_{t=1}^{T} q(\mathbf{x}_t|\mathbf{x}_{t-1})$. Hence, $\log\frac{q(\mathbf{h}|\boldsymbol{\eta},\mathcal{P})}{p_\theta(\boldsymbol{\eta},\mathbf{h}|\mathcal{P})}$ is actually

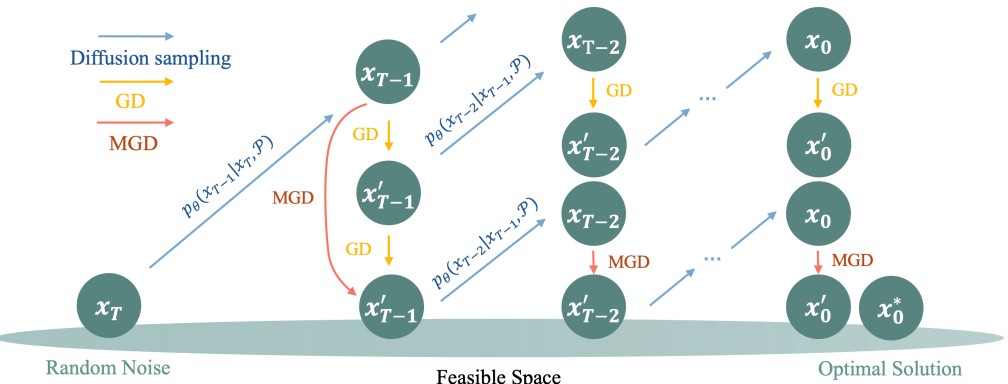

Figure 3: Visualization of objective-guided sampling with/without momentum.

the objective of diffusion models. It is noteworthy that, compared to other diffusion models, we need to further employ a solution decoder to transform the latent variables into the solution space. The objective guidance is provided from the original solution space instead of the feature space where the diffusion model is located. $\eta$ is initialized as the output of the diffusion model. We optimize $F$ concerning $\eta$ to learn a better intermediate variable. All the proposed solvers fit this framework.

**Gradient Descent with Momentum (MGD) Search** The aforementioned guidance method can be considered a special case of gradient descent, performing just a single optimization step on the diffusion latent variables. To enhance the effects of the guidance, we introduce momentum (Liu et al., 2020), a technique originally developed for neural network optimization, into the sampling. The intuition for momentum is to reduce the oscillation of the gradient updates and hence accelerate the optimization procedure. Since the objective-guided sampling process shares the same framework as model optimization, introducing momentum here can also enhance guidance. The update rule for this momentum is given by:

$$\mathbf{m} = \gamma \mathbf{m} - \varphi \mathbf{g}, \quad \mathbf{x} = \mathbf{x} + \mathbf{m} \tag{9}$$

where $\mathbf{g}$ represents the objective-guided gradients introduced previously. If we set $\gamma = 0$, then this update rule reduces to the original objective-guided formulation. The momentum mechanism in objective-guided sampling is visualized in Fig. 3. We can expect that with the introduction of momentum, the latent variables sampled reach feasibility faster compared to the original methods. If the gradient descent is performed only once for both sampling methods, the one with momentum bears less cumulative error and finds better solutions compared to the original sampling methods. The effectiveness of the momentum mechanism is proven through experiments.

## 4 EXPERIMENTS

### 4.1 BASELINES AND EVALUATION METRICS

We evaluate the following baselines on our datasets for better comparison and evaluation of the proposed methods: 1) Traditional solvers: Gurobi (Gurobi Optimization, LLC, 2024), and SCIP (Bolusani et al., 2024) and COPT (Ge et al., 2023). 2) Heuristic-based solvers: Relaxation Induced Neighbourhood Search (rins) (Danna et al., 2005) and feasibility pump (Fischetti et al., 2005). 3) Diffusion-based methods originally designed for binary ILP problems: IP Guided DDPM and DDIM (Zeng et al., 2024). For binary integer linear programming problems, we further compare with two other state-of-the-art methods: the Neural Diving (Nair et al., 2021), the Predict-and-Search algorithm (PS) (Nair et al., 2021) and DiffILO (Geng et al., 2025b).

We adopt four metrics: 1) Gap: the relative gap between the ground truth value and the predicted value, i.e. $\frac{|\mathbf{c}^\top \mathbf{x}_{gt} - \mathbf{c}^\top \mathbf{x}_{pred}|}{|\mathbf{c}^\top \mathbf{x}_{gt}|}$. The gap is only calculated among problems to which the solvers can get a feasible solution; 2) Time: the average computational time spent on evaluation. For generative models, the total time spent on all samples is recorded; 3) Sample feasibility: the average feasibility per instance. We choose to sample 30 times for all of the diffusion-based models to retrieve a high-quality solution. This metric can reflect how many of the samples can retrieve a feasible solution. Feasibility here means the satisfaction of the linear constraints, as the integrality constraints are enforced before evaluation through the hard rounding function. 4) Dataset feasibility: the average feasibility ratio of

Table 1: Results on classic binary integer linear programming. Fea: sample feasibility for generative models and dataset feasibility for non-generative models.

| Method | SC | | | CF | | | CA | | |
|---|---|---|---|---|---|---|---|---|---|
| | Gap | Time | Fea. | Gap | Time | Fea. | Gap | Time | Fea. |
| Gurobi (Gurobi Optimization, LLC, 2024) | 0.00% | 100s | 100% | 0.00% | 100s | 100% | 0.00% | 100s | 100% |
| SCIP (Bolusani et al., 2024) | 91.4% | 16.7m | 100% | 77.4% | 16.7m | 100% | 16.8% | 16.7m | 100% |
| rins (Danna et al., 2005) | 252.9% | 164.3s | 100% | NaN | 300.7s | 0.0% | 69.3% | 336.1s | 100% |
| feaspump (Fischetti et al., 2005) | 252.9% | 236.6s | 100% | 12.7% | 396.8s | 46.0% | 69.3% | 348.6s | 100% |
| PS (Han et al., 2023b) | 71.7% | 129.8s | 100% | 64.5% | 138.2s | 100% | 13.7% | 116.3s | 100% |
| Neural Diving (Nair et al., 2021) | NaN | 0.9s | 0.0% | NaN | 0.9s | 0.0% | 100% | 3.7s | 100% |
| Neural Diving+CompleteSol (Zeng et al., 2024) | 80.2% | 117.6s | 100% | 48.0% | 127.9s | 31.0% | 16.5% | 107.8s | 87.0% |
| IP Guided DDPM (Zeng et al., 2024) | 70.8% | 11h | 95.7% | 80.5% | 30h | 44.0% | 98.6% | 9h | 100% |
| IP Guided DDIM (Zeng et al., 2024) | 68.5% | 65m | 99.8% | 54.6% | 1.5h | 89.7% | 25.4% | 77m | 97.1% |
| DiffILO (Geng et al., 2025b) | 93.9% | 22.2s | 100% | 512.3% | 15.2s | 100% | 99.2% | 33.2s | 100% |
| CMILP (Ours) | 90.2% | 21.7s | 100% | 79.2% | 2.3m | 92.1% | 80.2% | 51.1s | 100% |
| SCMILP (Ours) | 91.6% | 27.2s | 100% | 82.9% | 2.9m | 88.3% | 85.3% | 36.1s | 100% |
| MFILP (Ours) | 88.4% | 21.3s | 100% | 76.1% | 2.3m | 89.7% | 79.2% | 32.8s | 100% |

the dataset. This metric reflects the percentage of problems in which the solvers can find a feasible solution. Dataset feasibility is a more commonly evaluated metric compared to sample feasibility. Sample feasibility and dataset feasibility can reflect the performance of generative-model-based solvers from different perspectives.

## 4.2 BINARY INTEGER LINEAR PROGRAMMING PROBLEMS

In this section, we assess our methods' capacity on three classic binary integer linear programming (BILP) problems, i.e., set cover, capacitated facility location, and combinatorial auction. All variables in the problems are binary variables. The instances are all generated by the Ecole library (Prouvost et al., 2020). Given the high complexity of these problems, we adopt solutions obtained by Gurobi with a 100-second time limit as training targets. The training dataset consists of 800 instances, while the test set contains 100 instances. For evaluation, SCIP is run with a 1000-second limit to obtain suboptimal solutions. PS leverages Gurobi as the post-processor and follows parameters settings used by Han et al. (2023b). For neural diving (Nair et al., 2021), we use a low-coverage (coverage=0.2) model that emphasizes solution feasibility to complete partial solutions. We also report results on neural diving with the CompleteSol heuristics from SCIP (Bolusani et al., 2024) as the post-processing techniques.

The experimental results are summarized in Table 1. Since all diffusion-based models achieve 100% dataset feasibility across all datasets, we report only the remaining three metrics in the table. As shown, our method attains higher sample feasibility than both IP Guided DDPM and DDIM. Additionally, on the CF and CA datasets, our approach achieves a smaller optimality gap than IP Guided DDPM while requiring less inference time. Although IP Guided DDIM consistently produces the lowest gap across all datasets, its inference time is considerably longer compared to both our method and traditional solvers.

## 4.3 NON-BINARY INTEGER LINEAR PROGRAMMING PROBLEMS

### 4.3.1 INVENTORY MANAGEMENT DATASETS

In this section, we perform experiments on non-binary linear programming problems. We mainly focus on two artificial datasets. The first dataset tries to model the inventory management problems. We could form the problems as:

$$\min \sum_{i=1}^{m} \sum_{j=1}^{n} s_j x_{ij} \quad \text{s.t.} \sum_{j=1}^{n} x_{ij} \geq q_i, \sum_{i=1}^{m} a_i x_{ij} \leq C_j, x_{ij} \geq 0, x_{ij} \in \mathbb{Z} \tag{10}$$

The inventory problems aim to minimize the inventory costs while ensuring that the storage satisfies the demands and that the total storage in need doesn't exceed the storage space. For simplicity, we also add an upper limit on the number of each single type of goods that each warehouse could purchase. All coefficients were generated by sampling integer values uniformly from an interval. We can hence define an inventory management problem as IM-(n, m, b), where n is the number of warehouses, m is the number of types of goods, and b is the variable upper bound. IM-(n, m, b) has $n \times m$ variables and $m + n$ constraints. We generate 800 instances for the training dataset and 100 for the testing dataset. The instances are labeled by Gurobi.

Table 2: Experimental results on small-scale inventory management datasets where the number of warehouses is larger than the number of types of goods. S. Fea is the abbreviation of sample feasibility, and D. Fea is the abbreviation of dataset feasibility.

| Method | IM-(50, 5, 2) | | | | IM-(50, 5, 5) | | | | IM-(50, 5, 10) | | | |
|---|---|---|---|---|---|---|---|---|---|---|---|---|
| | Gap | Time | S. Fea. | D. Fea. | Gap | Time | S. Fea. | D. Fea. | Gap | Time | S. Fea. | D. Fea. |
| Gurobi (Gurobi Optimization, LLC, 2024) | 0.00% | 6.6s | - | 100% | 0.00% | 4.6s | - | 100% | 0.00% | 5.8m | - | 100% |
| SCIP (Bolusani et al., 2024) | 0.00% | 13.2s | - | 100% | 0.00% | 6.7s | - | 100% | 0.00% | 28h | - | 100% |
| COPT (Ge et al., 2023) | 0.00% | 32.2s | - | 100% | 0.00% | 31.8s | - | 100% | 0.00% | 17.9m | - | 100% |
| rins (Danna et al., 2005) | 0.61% | 4.2s | - | 61% | 0.00% | 3.6% | - | 54.0% | 0.00% | 4.7s | - | 6.0% |
| feaspump (Fischetti et al., 2005) | 0.62% | 3.4s | - | 60.0% | 0.00% | 2.7s | - | 53.0% | 0.00% | 3.7s | - | 6.0% |
| Neural Diving (Nair et al., 2021) | NaN | 0.7s | - | 0.0% | NaN | 0.7s | - | 0.0% | NaN | 0.8s | - | 0.0% |
| Neural Diving+CompleteSol (Zeng et al., 2024) | 21.2% | 3.1s | - | 28.0% | 21.3% | 3.3s | - | 61.0% | 57.3% | 5.0s | - | 72.0% |
| IP Guided DDPM (Zeng et al., 2024) | 92.9% | 34m | 0.1% | 13.0% | 15.6% | 48m | 0.1% | 13.0% | 87.2% | 28m | 0.1% | 1.0% |
| IP Guided DDIM (Zeng et al., 2024) | 15.0% | 6m | 46.0% | 80.0% | 6.0% | 5m | 32.3% | 88.0% | 133.3% | 7.3m | 18.6% | 68.0% |
| CMILP (Ours) | 16.5% | 2.6s | 69.2% | 88.0% | 8.4% | 2.8s | 71.3% | 90.0% | 119.3% | 3.0s | 35.7% | 76.0% |
| SCMILP (Ours) | 12.2% | 2.0s | 42.4% | 78.0% | 10.1% | 2.3s | 35.8% | 86.0% | 112.9% | 2.9s | 20.3% | 62.0% |
| MFILP (Ours) | 12.1% | 2.1s | 70.5% | 90.0% | 11.4% | 2.0s | 60.6% | 80.0% | 107.1% | 2.1s | 36.8% | 68.0% |

Table 3: Experimental results on inventory management datasets

| Method | IM-(5, 50, 2) | | | | IM-(200, 5, 2) | | | | IM-(100, 10, 2) | | | |
|---|---|---|---|---|---|---|---|---|---|---|---|---|
| | Gap | Time | S. Fea. | D. Fea. | Gap | Time | S. Fea. | D. Fea. | Gap | Time | S. Fea. | D. Fea. |
| Gurobi (Gurobi Optimization, LLC, 2024) | 0.00% | 48.3s | - | 100% | 0.00% | 46.6s | - | 100% | 0.00% | 53.3s | - | 100% |
| SCIP (Bolusani et al., 2024) | 0.00% | 29.1s | - | 100% | 0.00% | 80.87s | - | 100% | 0.00% | 8h | - | 100% |
| COPT (Ge et al., 2023) | 0.00% | 4.9m | - | 100% | 0.00% | 38.5s | - | 100% | 0.00% | 4.2m | - | 100% |
| rins (Danna et al., 2005) | 0.00% | 1.8s | - | 71.0% | 0.00% | 15.6s | - | 42.0% | 0.0% | 22.1s | - | 3.0% |
| feaspump (Fischetti et al., 2005) | 0.00% | 1.5s | - | 88.0% | 0.00% | 13.9s | - | 43.0% | NaN | 20.8s | - | 0.0% |
| Neural Diving (Nair et al., 2021) | NaN | 0.8s | - | 0.0% | NaN | 0.8s | - | 0.0% | NaN | 0.7s | - | 0.0% |
| Neural Diving+CompleteSol (Zeng et al., 2024) | NaN | 2.9s | - | 0.0% | 21.7% | 5.9s | - | 7.0% | 20.4% | 5.8s | - | 7.0% |
| IP Guided DDPM (Zeng et al., 2024) | 61.2% | 39m | 0.1% | 1.0% | 109.1% | 1.7h | 3.3% | 1.0% | 21.2% | 2h | 0.9% | 16.0% |
| IP Guided DDIM (Zeng et al., 2024) | 6.6% | 14m | 73.3% | 92.0% | 10.2% | 36m | 60.5% | 89.0% | 13.2% | 42m | 35.0% | 76.0% |
| CMILP (Ours) | 4.9% | 1.9s | 52.8% | 89.0% | 10.8% | 17.0s | 79.4% | 90.0% | 18.0% | 18.6s | 36.3% | 67.0% |
| SCMILP (Ours) | 5.3% | 2.2s | 67.3% | 88.0% | 15.8% | 23.6s | 42.8% | 86.0% | 17.5% | 26.4s | 15.6% | 62.0% |
| MFILP (Ours) | 5.7% | 1.9s | 54.3% | 80.0% | 9.2% | 19.2s | 71.3% | 90.0% | 16.1% | 19.2s | 37.7% | 69.0% |

Table 4: Experimental results on inventory management datasets and their binarized variants

| Method | IM-(50, 5, 2) | | | | Binarized IM-(50, 5, 2) | | | | IM-(50, 5, 5) | | | | Binarized IM-(50, 5, 5) | | | |
|---|---|---|---|---|---|---|---|---|---|---|---|---|---|---|---|---|
| | Gap | Time | S. Fea. | D. Fea. | Gap | Time | S. Fea. | D. Fea. | Gap | Time | S. Fea. | D. Fea. | Gap | Time | S. Fea. | D. Fea. |
| IP Guided DDPM | 92.9% | 34m | 0.1% | 1.0% | NaN | 101m | 0.0% | 0.0% | 15.6% | 48m | 0.1% | 13.0% | 79.6% | 1.7h | 1.7% | 15.0% |
| IP Guided DDIM | 15.0% | 6m | 46.0% | 80.0% | NaN | 19m | 0.0% | 0.0% | 6.0% | 5m | 32.3% | 88.0% | 32.6% | 18.5m | 25.6% | 53.0% |
| CMILP (Ours) | 16.5% | 2.6s | 69.2% | 88.0% | 0.0% | 12.2s | 0.6% | 3.0% | 8.4% | 2.8s | 71.3% | 90.0% | 0.0% | 9.8s | 2.1% | 8.0% |
| SCMILP (Ours) | 12.2% | 2.0s | 42.4% | 78.0% | 0.0% | 17.2s | 0.3% | 3.0% | 10.1% | 2.3s | 35.8% | 86.0% | 4.4% | 10.1s | 0.3% | 5.0% |
| MFILP (Ours) | 12.1% | 2.1s | 70.5% | 90.0% | 0.0% | 13.4s | 0.3% | 3.0% | 11.4% | 2.0s | 60.6% | 80.0% | 2.8% | 9.8s | 1.2% | 9.0% |

Experiment results are shown in Table 2 and Table 3. In Table 2, we present experiment results on relatively small-scale instances where the number of warehouses is larger than the number of types of goods. It could be observed that the proposed one-step diffusion solvers find solutions faster compared to traditional solvers. Our models achieve comparative performance on gap, sample feasibility, and dataset feasibility in far less time than IP Guided DDPM and DDIM.

In Table 3, we examine models' performance on inventory management problems where the number of types of goods exceeds the number of warehouses and larger-scale datasets. Overall performance trends remain consistent with Table 2. While IP Guided DDIM achieves higher dataset feasibility on IM-(5, 50, 2), it suffers from significantly longer solving times and larger optimality gaps.

In Table 4, we compare the models' performance on the vanilla form that we used and the binarized variant commonly adopted in literature. Binarization significantly increases problem size and solving time. For example, IM-(50, 5, 5) is a dataset with variables taking 6 distinct integer values. If we use a binary variable transformation to turn the dataset into a binary ILP instance, the problem will be turned into an optimization problem with more than 1000 variables. Table 4 confirms that binarization imposes additional computational burdens on neural solvers. Our introduction of the IIP layer helps address this issue by maintaining problem compactness and improving model performance without the need for costly variable transformations.

Finally, we evaluate the newly devised gradient descent with momentum (MGD) search methods on the most complicated dataset, IM-(50, 5, 10). The wide bound of variables makes it hard for the solvers to achieve satisfactory results. The results are shown in Table 5. It could be concluded that the introduction of momentum

Table 5: Experimental results on IM-(50, 5, 10) with different gradient search schemes. $T_i$ stands for the number of model inference steps.

| Method | IM-(50, 5, 10) | | | |
|---|---|---|---|---|
| | Gap | Time | S. Fea. | D. Fea. |
| SCMILP ($T_i = 10$, Opt=GD) | 104.5% | 22.9s | 29.5% | 78.0% |
| SCMILP ($T_i = 10$, Opt=MGD) | 101.8% | 24.9s | 30.3% | 82.0% |
| SCMILP ($T_i = 20$, Opt=GD) | 99.8% | 32.5s | 35.1% | 87.0% |
| SCMILP ($T_i = 20$, Opt=MGD) | 95.8% | 36.6s | 35.5% | 88.0% |

improves the search quality significantly while generally maintaining the solving time unchanged. The momentum mechanism raises the dataset feasibility by as much as 4% and reduces the gap by

Table 6: Experimental results on synthetic non-binary ILP datasets.

| Method | Random-(500, 20, 2) | | | | Random-(1000, 20, 2) | | | | Random-(2000, 20, 2) | | | |
|---|---|---|---|---|---|---|---|---|---|---|---|---|
| | Gap | Time | S. Fea. | D. Fea. | Gap | Time | S. Fea. | D. Fea. | Gap | Time | S. Fea. | D. Fea. |
| Gurobi (Gurobi Optimization, LLC, 2024) | 0.00% | 5.4s | - | 100% | 0.00% | 18.1s | - | 100% | 0.00% | 4.2s | - | 100% |
| SCIP (Bolusani et al., 2024) | 0.00% | 9.2s | - | 100% | 0.00% | 27.6s | - | 100% | 0.00% | 48.4s | - | 100% |
| COPT (Ge et al., 2023) | 0.00% | 36.3s | - | 100% | 0.00% | 40.7s | - | 100% | 0.00% | 46.7s | - | 100% |
| rins (Danna et al., 2005) | 0.00% | 7.1s | - | 41.0% | 0.00% | 10.8s | - | 31.0% | 0.00% | 22.4s | - | 14.0% |
| feaspump (Fischetti et al., 2005) | 0.61% | 5.5s | - | 70.0% | 0.30% | 9.3s | - | 82.0% | 2.05% | 21.2s | - | 72.0% |
| Neural Diving (Nair et al., 2021) | NaN | 0.8s | - | 0.0% | NaN | 2.8s | - | 0.0% | NaN | 3.1s | - | 0.0% |
| Neural Diving+CompleteSol (Zeng et al., 2024) | 21.9% | 5.1s | - | 100% | 22.6% | 6.9s | - | 100% | 99.4% | 11.8s | - | 97.0% |
| IP Guided DDPM (Zeng et al., 2024) | 10.3% | 1.2h | 43.4% | 100% | 1.2% | 1.9h | 3.0% | 22.0% | 0.5% | 4h | 7.3% | 71.0% |
| IP Guided DDIM (Zeng et al., 2024) | 0.7% | 14m | 85.1% | 100% | 0.3% | 20m | 77.1% | 96.0% | 0.3% | 46m | 9.26% | 70.0% |
| CMILP (Ours) | 0.0% | 3.1s | 46.8% | 85.0% | 0.5% | 9.7s | 16.3% | 87.0% | 1.1% | 21.2s | 14.5% | 75.0% |
| SCMILP (Ours) | 0.2% | 4.4s | 42.0% | 88.0% | 0.0% | 10.3s | 37.7% | 89.0% | 0.3% | 22.2s | 14.8% | 74.0% |
| MFILP (Ours) | 0.0% | 3.6s | 45.4% | 82.0% | 0.0% | 7.1s | 26.7% | 85.0% | 0.0% | 19.4s | 11.7% | 85.0% |

roughly 2%. Further, with the increasing number of inference steps, we can see that the performance of the shortcut model rises steadily. We can change the number of steps according to the requirements of the application scenarios, making our methods more applicable in real-life settings.

### 4.3.2 SYNTHETIC NON-BINARY INTEGER LINEAR PROGRAMMING DATASETS

The inventory management problem is a special type of integer linear programming problem. To further examine our models' performance, we generate a set of synthetic non-binary ILP datasets in the form of Eq. 1. We adopt the instance generation procedure introduced by Lee & Kim (2025), where the generated problems are guaranteed to be bounded and feasible. Each coefficient is drawn from a discrete uniform distribution over the integer range. For simplicity, we also add a variable upper bound. We term a dataset with n variables, m constraints, and a variable bound of b as Random-(n, m, b). As in inventory management datasets, we generate 800 instances for the training dataset and 100 for the testing dataset. The instances are labeled by Gurobi.

Table 6 reports results on larger-scale synthetic datasets. Interestingly, despite the increased problem size, traditional solvers exhibit shorter solving times, as seen in Random-(500, 20, 2). This occurs because problem difficulty is not fully captured by the number of variables and constraints alone. In contrast, neural solvers show increased inference time proportional to the problem dimensions, as their computational overhead is primarily governed by the variable and constraint counts, which puts IP Guided DDPM and DDIM at a relative disadvantage. Our models, however, can accurately solve most instances in significantly less time than Gurobi and SCIP. Moreover, in terms of solution quality, a few additional steps allow our models to achieve comparable performance—for example, on Random-(1000, 20, 2), it requires 5 steps and 57 seconds.

### 5 CONCLUSION AND LIMITATIONS

This paper presents three one-step, end-to-end diffusion solvers—CMILP, SCMILP, and MFILP—that generate feasible solutions for general integer linear programming problems, a domain that has been largely unexplored due to its inherent complexity. To extend ILP neural solvers to general instances, we introduce a novel iterative integer projection (IIP) layer. Additionally, we integrate a momentum mechanism into the objective-guided sampling of diffusion models to enhance solution guidance. Experimental results demonstrate the superiority of our methods in both runtime and solution quality. Limitations include a relatively big optimality gap compared to traditional solvers, and the computational cost of gradient-based search increases substantially with dataset size—a challenge common to all loss-guided diffusion approaches.

### ETHICS STATEMENT

This paper aims to advance the state of the art in learning integer linear programming. While the research may entail various societal implications, we do not identify any that warrant specific emphasis in this paper.

### REPRODUCIBILITY STATEMENT

All experimental results in the paper are reproducible, and the implementation code for reproducing experimental results will be fully open sourced on Github after the paper is accepted.

## LLM USAGE STATEMENT

The contribution of LLM in the work proposed in this article is limited to: 1. polishing given written statements; 2. Given written sentence syntax review. We declare that no experimental data was generated/modified by LLM.

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

Table 7: Experimental results on smaller-scale synthetic non-binary ILP Datasets

| Method | Random-(300, 30, 2) | | | | Random-(300, 20, 5) | | | |
|---|---|---|---|---|---|---|---|---|
| | Gap | Time | S. Fea. | D. Fea. | Gap | Time | S. Fea. | D. Fea. |
| Gurobi Gurobi Optimization, LLC (2024) | 0.00% | 14.0s | - | 100% | 0.00% | 12.3s | - | 100% |
| SCIP Bolusani et al. (2024) | 0.00% | 11.1s | - | 100% | 0.00% | 18.2s | - | 100% |
| rins Danna et al. (2005) | 4.6% | 10.7s | - | 16.0% | 0.0% | 4.9s | - | 19.0% |
| feaspump Fischetti et al. (2005) | 5.1% | 11.9s | - | 31.0% | 0.8% | 5.5s | - | 37.0% |
| IP Guided DDPM Zeng et al. (2024) | 11.3% | 26m | 0.9% | 16.0% | 15.9% | 23m | 2.3% | 40.0% |
| IP Guided DDIM Zeng et al. (2024) | 17.5% | 8m | 50.8% | 88.0% | 2.2% | 5m | 82.7% | 97.0% |
| CMILP (Ours) | 0.0% | 1.8s | 45.5% | 62.0% | 0.3% | 1.9s | 16.9% | 78.0% |
| SCMILP (Ours) | 0.2% | 2.3s | 8.5% | 52.0% | 0.1% | 2.8s | 27.5% | 70.0% |
| MFILP (Ours) | 0.0% | 1.7s | 43.4% | 59.0% | 0.1% | 1.7s | 26.7% | 77.0% |

Table 8: Experimental results on small-scale inventory management datasets with different penalty coefficient.

| Method | IM-(50, 5, 2) | | | | IM-(50, 5, 5) | | | | IM-(50, 5, 10) | | | |
|---|---|---|---|---|---|---|---|---|---|---|---|---|
| | Gap | Time | S. Fea. | D. Fea. | Gap | Time | S. Fea. | D. Fea. | Gap | Time | S. Fea. | D. Fea. |
| IP Guided DDPM (Zeng et al., 2024) | 92.9% | 34m | 0.1% | 1.0% | 15.6% | 48m | 0.1% | 13.0% | 87.2% | 28m | 0.1% | 1.0% |
| IP Guided DDPM(penalty coef=0) (Zeng et al., 2024) | NaN | 34m | 0.0% | 0.0% | NaN | 48m | 0.0% | 0.0% | NaN | 28m | 0.0% | 0.0% |
| IP Guided DDIM (Zeng et al., 2024) | 15.0% | 6m | 46.0% | 80.0% | 6.0% | 5m | 32.3% | 88.0% | 133.3% | 7.3m | 18.6% | 68.0% |
| IP Guided DDIM (penalty coef=0) (Zeng et al., 2024) | NaN | 6m | 0.0% | 0.0% | NaN | 5m | 0.0% | 0.0% | NaN | 7.3m | 0.0% | 0.0% |
| CMILP (Ours) | 16.5% | 2.6s | 69.2% | 88.0% | 8.4% | 2.8s | 71.3% | 90.0% | 119.3% | 3.0s | 35.7% | 76.0% |
| CMILP (penalty coef=0) (Ours) | NaN | 2.6s | 0.0% | 0.0% | NaN | 2.8s | 0.0% | 0.0% | NaN | 3.0s | 0.0% | 0.0% |
| SCMILP (Ours) | 12.2% | 2.0s | 42.4% | 78.0% | 10.1% | 2.3s | 35.8% | 86.0% | 112.9% | 2.9s | 20.3% | 62.0% |
| SCMILP (penalty coef=0) (Ours) | NaN | 2.0s | 0.0% | 0.0% | NaN | 2.3s | 0.0% | 0.0% | NaN | 2.9s | 0.0% | 0.0% |
| MFILP (Ours) | 12.1% | 2.1s | 70.5% | 90.0% | 11.4% | 2.0s | 60.6% | 80.0% | 107.1% | 2.1s | 36.8% | 68.0% |
| MFILP (penalty coef=0) (Ours) | NaN | 2.1s | 0.0% | 0.0% | NaN | 2.0s | 0.0% | 0.0% | NaN | 2.1s | 0.0% | 0.0% |

## A    ADDITIONAL RESULTS

We test our model on small-scale, randomly generated datasets. The results are shown in Table 7. CMILP performs the best on Random-(300, 30, 2). It takes only 1.82 seconds to finish solving, while for IP Guided DDPM and DDIM, the solving procedure generally takes minutes. Hence, on general integer linear programming problems, our models are still more practical compared to IP Guided DDPM and DDIM. On Random-(300, 20, 5), although IP Guided DDIM achieves the highest dataset feasibility, its gap and solving time are way too high compared to our models. To achieve comparable dataset feasibility, it takes CMILP 20 steps and 30 seconds and takes Shortcut 2 steps and 4 seconds. Generally, Shortcut beats IP Guided DDIM on this dataset, further showcasing our models' capacity on general integer linear programming problems. Furthermore, if we turn Random-(300, 20, 5) into binary ILP problems, it generally takes 10 times longer time to finish solving, as can be inferred from datasets of similar sizes as in Table 6. This will waste the speed advantage of neural-network-based solvers. We should always try to tackle integer linear programming problems directly instead of converting those problems to the binary versions.

## B    ANALYSIS ON THE FEASIBILITY PENALTY

In this section, we attempt to analyze effectiveness of the feasibility penalty. Constraint satisfaction is one key factor when evaluating the ILP solvers. The feasibility penalty is introduced to enforce constraint satisfaction more effectively. The results are shown in Table 8. We can infer from the table that neural solvers trained without the feasibility penalty can't generate feasible solutions at all. Our introduction of the feasibility penalty successfully enhance the models' performance.

## C    SCMILP: SHORTCUT DIFFUSION MODEL FOR INTEGER LINEAR PROGRAMMING

Recently there has been a new variant of the diffusion model that can also generate high-quality solutions in one or a few steps, the shortcut model (Frans et al., 2024). The shortcut model is built upon a flow matching model (Lipman et al., 2023). The flow matching model attempts to learn a

vector field that transports a random Gaussian distribution to the target distribution. The original flow matching models suffer from the large number of inference steps required to generate a high-quality solution, as with the diffusion model. The shortcut model tackles this issue by conditioning not only on the problem instance but also on the size of the inference steps. The shortcut $s_\theta(\mathbf{x}_t, t, d)$, which is defined as the normalized direction to the next variable, is hence introduced:

$$\mathbf{x}'_{t+d} = \mathbf{x}_t + s_\theta(\mathbf{x}_t, t, d)d \tag{11}$$

where $d$ is the step length, $s_\theta(\mathbf{x}_t, t, d)$ represents the velocity we take at state $\mathbf{x}_t$ given a time step of size $d$. The shortcut model is trained using a combination of the self-consistency loss and the flow matching loss. The self-consistency refers to the model quality that one shortcut step equals two consecutive shortcut steps of half the size. The flow matching loss tries to supervise with the ground truth vector field. This loss enables the model to function under large sampling steps.

$$\mathcal{L}^{N_{t,d}}_{\text{SCMILP}}(\theta) = \mathbb{E}\left[\|s_\theta(\mathbf{x}_t, t, 0) - (\mathbf{x}_1 - \mathbf{x}_0)\| + \|s_\theta(\mathbf{x}_t, t, 2d) - \frac{s_\theta(\mathbf{x}_t, t, d) + s_\theta(\mathbf{x}'_{t+d}, t, d)}{2}\|\right] \tag{12}$$

where $(t, d)$ is sampled according to the time scheduler $N_{t,d}$. For the shortcut model, the step size $d$ is embedded using the sinusoidal embedding and together with the time $t$ as the conditional inputs.

## D MFILP: MEANFLOW MODEL FOR INTEGER LINEAR PROGRAMMING

The mean flow model (Geng et al., 2025a) is another generative model that instead uses the average velocity $u(\mathbf{x}_t, r, t)$, where $[r, t]$ is the time window of the average velocity, to capture distributional changes, in contrast to the instantaneous velocity $v(\mathbf{x}_t, t)$ modeled in flow matching. An identity relationship forms between those two velocities:

$$u(\mathbf{x}_t, r, t) = v(\mathbf{x}_t, t) - (t - r)\frac{d}{dt}u(\mathbf{x}_t, r, t), \text{ where } \frac{d}{dt}u(\mathbf{x}_t, r, t) = v(\mathbf{x}_t, t)\partial_x u + \partial_t u \tag{13}$$

As in flow matching, the instantaneous velocity $v(\mathbf{x}_t, t)$ is modeled as $v(\mathbf{x}_t, t) = \epsilon - \mathbf{x}_t$. Eq. 13 can hence provide the target average velocity in arbitrary time ranges $[r, t]$. The neural network is trained to approximate this average velocity by minimizing the following loss with the time scheduler $N_{r,t}$.

$$\mathcal{L}^{N_{r,t}}_{\text{MFILP}} = \mathbb{E}\|u_\theta(\mathbf{x}_t, r, t) - u_{\text{target}}\|_2^2 \tag{14}$$

