# OpenReview forum: "One-Step Diffusion Solver for Non-binary Integer Linear Programming"
_ICLR.cc/2026/Conference — Submitted to ICLR 2026_

### Official Review · Reviewer_qDWz · 2025-10-30

**Soundness:** 2
**Presentation:** 2
**Contribution:** 3
**Rating:** 4
**Confidence:** 4

**Summary:**

This paper presents three one-step diffusion-based solvers—CMILP, SCMILP, and MFILP—for solving mixed-integer linear programming (MILP) problems. Inspired by consistency, shortcut, and meanflow training techniques, these solvers incorporate a novel Iterative Integer Projection (IIP) layer to directly handle non-binary integer constraints, eliminating the need for costly problem transformations. Experimental results show that the proposed approach outperforms existing learning-based methods on both binary and non-binary ILP instances, while demonstrating stronger scalability compared to traditional solvers.

**Strengths:**

1.  Leveraging diffusion models for MILP solution prediction is a timely and effective choice. Diffusion models offer greater expressive power than graph neural networks (GNNs), enabling more accurate capture of complex solution distributions for ILP problems.

2. The proposed Iterative Integer Projection (IIP) layer addresses the gap in existing literature, which predominantly focuses on binary integer programming. By enabling direct handling of non-binary integer constraints, the method avoids exponential problem size growth associated with binary transformation, enhancing efficiency for general ILP scenarios.

**Weaknesses:**

1. Incomplete baseline comparisons: Key solution prediction baselines are omitted, including Contrastive Predict-and-Search and FMILP. This omission limits the comprehensiveness of the performance evaluation, as these methods are widely recognized in the field of learning-based ILP solvers. The authors may want to refer to the following related works in the paper.
[1] Apollo-MILP: An Alternating Prediction-Correction Neural Solving Framework for Mixed-Integer Linear Programming. ICLR, 2025
[2] FMIP: Joint Continuous-Integer Flow For Mixed-Integer Linear Programming.
[3] Effective Generation of Feasible Solutions for Integer Programming via Guided Diffusion, KDD, 2024.

2. Limited benchmark: The paper does not clearly specify the full scope of the dataset sizes. Additionally, the instances appear relatively easy to solve—evidenced by Gurobi’s ability to find optimal solutions within 100 seconds in Tables 1, 2, and 3. Evaluations on larger-scale instances or real-world benchmarks are needed to validate the method’s scalability and practical utility fully.

3. The proposed solvers exhibit relatively large optimality gaps compared to traditional solvers and some advanced learning-based methods. While inference time is short, the balance between speed and solution quality (including feasibility) is not fully optimized, which may restrict applicability in scenarios requiring high-precision solutions.

4. Insufficient implementation details: Critical implementation specifics are unclear, such as whether PS/Neural Diving were used for neighborhood search and the key hyperparameters adopted for PS and DiffILO. Notably, the paper reports results where PS performs worse than Gurobi, which contradicts the findings of PS’s original study—even on datasets presumably consistent with the original work. This discrepancy raises questions about the experimental setup and requires clarification.

**Questions:**

1. Why do the authors not use the same problem sizes used in the Predict-and-Search paper?

---

> ### Author Response · Authors · 2025-11-22
>
> **Q1: Incomplete baseline comparisons and limited benchmark**
>
> A1: Thank you very much for raising this concern. Firstly, regarding baseline, since FMIP's major innovation is for its combination of continuous and discrete flow matching and our datasets in the paper are mainly only integer programming problems, we didn't consider it as a major baseline to be compared with. As regards predict and search methods, we have already included in the binary integer linear programming problems.
>
> Secondly, regarding the limited benchmark scope: in response to the reviewer’s suggestion, we added experiments on a mixed-integer benchmark—the item placement problem—to better illustrate the applicability of our method to MILPs. Following FMIP, we adopt a tri-partite graph encoder to extract problem features. However, due to the complexity of this MILP, feasibility requires post-processing. Notably, even using the official checkpoints provided by FMIP, we were unable to obtain feasible solutions without such post-processing. Therefore, comparing all methods under the same post-processing protocol is the fairest and most meaningful evaluation.
>
> While our model performs slightly worse than FMIP on this benchmark, it is important to highlight that our model is significantly more lightweight. FMIP relies on a 12-layer tri-partite GCN to model the diffusion process, whereas our approach uses the GCN only as an initial feature extractor and applies a transformer-based module for diffusion. Transformers are substantially more efficient than deep GCNs, and our method requires only a single inference step, demonstrating clear computational advantages.
>
> Finally, we emphasize that the central innovation of our work lies in its ability to handle non-binary integer linear programs effectively. The item placement problem used in this benchmark remains binary with respect to integer variables, and there currently exist very few open-source generators for non-binary MILPs. Expanding our evaluation to more challenging non-binary MILPs is an important direction that we plan to pursue in future work.
>
> |                | Pred&Search (600s)     | Apollo (800s)          | Neural Diving (400s)   |
> | -------------- | ---------------------- | ---------------------- | ---------------------- |
> | SL             | Gap -13.48%, Obj 15.34 | Gap -21.13%, Obj 14.16 | Gap -17.54%, Obj 14.62 |
> | FMIP (Time 20.12s per instance)          | Gap -21.48%, Obj 13.92 | Gap -22.50%, Obj 13.74 | Gap -21.09%, Obj 13.99 |
> | IP-Guided-Diff | Gap -14.21%, Obj 15.21 | Gap -19.17%, Obj 14.33 | Gap -17.48%, Obj 14.63 |
> |MFILP (Ours, 0.62s per instance)|Gap -21.13%, Obj 14.16|Gap -21.48%, Obj 13.82|Gap -19.97%, Obj 14.33|

---

> > ### Author Response · Authors · 2025-11-22
> >
> > **Q2: The proposed solvers exhibit relatively large optimality gaps compared to traditional solvers and some advanced learning-based methods. While inference time is short, the balance between speed and solution quality (including feasibility) is not fully optimized, which may restrict applicability in scenarios requiring high-precision solutions.**
> >
> > A2: Thank you very much for raising this concern. To address it, we additionally provide results from traditional exact solvers under comparable time constraints. For a fair comparison, we use the time required to generate a single sample with our model as the time limit for each exact solver. The results show that, under the same time budget, our method achieves competitive performance relative to these solvers. This further highlights the effectiveness of our approach in tackling non-binary integer linear programming problems, especially in settings where strict time limits are imposed.
> >
> > Second, we wish to emphasize that our primary comparison target is neural solvers, rather than exact solvers. At present, neural methods generally cannot match the exactness of traditional solvers without additional heuristics or specialized decoding strategies. In this context, our method demonstrates clear improvements: compared with existing neural approaches, it achieves higher feasibility rates and lower optimality gaps. We believe this represents a meaningful advancement in the development of neural solvers for ILP and MILP.
> >
> > Looking forward, we plan to explore enhanced neural architectures and algorithmic designs that may ultimately enable neural solvers to close the remaining gap with traditional optimization solvers.
> >
> > |         | IM-(200, 5, 2)                     | IM-(100, 10, 2)                    | Random-(1000, 20, 2)|
> > | ------ | -------------------------------- | ---------------------------------- | ---------------------------------- |
> > | Gurobi  | Gap 0.0%, Time 46.6s, D. Feas 100% | Gap 0.0%, Time 53.3s, D. Feas 100% | Gap 0.0%, Time 18.1s, D. Feas 100%|
> > | Gurobi with limit  | Gap 12.1%, Time 3.9s, D. Feas 95% | Gap 13.5%, Time 4.2s, D. Feas 95% | Gap 2.2%, Time 2.6s, D. Feas 100% |
> > | COPT with limit | Gap 99.0%, Time 4.0s, D. Feas 100% | Gap 100%, Time 4.7s, D. Feas 100% | Gap 1e22%, Time 2.9s, D. Feas 100% |
> > | CMILP (1 sample time) | Gap 10.8%, Time 3.9s, D. Feas 90% | Gap 18.0%, Time 4.6s, D. Feas 67% | Gap 0.5%, Time 2.8s, D. Feas 87% |
> > | SCMILP (1 sample time) |  Gap 15.8%, Time 5.4s, D. Feas 86% | Gap 17.5%, Time 4.2s, D. Feas 62% | Gap 0.0%, Time 3.2s, D. Feas 89%  |
> > | MFILP (1 sample time)  | Gap 9.2%, Time 4.7s, D. Feas 90%  | Gap 16.1%, Time 4.6s, D. Feas 69% | Gap 0.0%, Time 2.6s, D. Feas 85% |
> >
> >
> >
> > **Q3: Insufficient implementation details: Critical implementation specifics are unclear, such as whether PS/Neural Diving were used for neighborhood search and the key hyperparameters adopted for PS and DiffILO. Notably, the paper reports results where PS performs worse than Gurobi, which contradicts the findings of PS’s original study—even on datasets presumably consistent with the original work. Why do the authors not use the same problem sizes used in the Predict-and-Search paper?**
> >
> > A3: Thank you for raising this concern. We follow the same experimental settings as DiffILO. However, DiffILO relies on SCIP as a post-processing step, whereas our goal is to develop a fully end-to-end integer programming (IP) solver. Therefore, we remove the SCIP post-processing to evaluate the raw performance of the model itself. This difference in evaluation protocol is the main reason for the discrepancy from the numbers reported in the original DiffILO paper.
> >
> > Regarding Predict-and-Search (PS), we directly cite the results presented in [1]. In their setup, Gurobi is used as a heuristic solver to quickly extract feasible solutions, rather than as an exact solver. This difference in solver configuration can also explain the numerical mismatch with the original PS paper, where Gurobi was run in exact mode.
> >
> > Finally, we adopt the same problem sizes as in the original PS paper. However, we do not include all datasets used by PS. Our focus is specifically on non-binary integer linear programming problems, where existing neural solvers—most designed for binary MILPs—struggle. Our introduction of the IIP formulation significantly accelerates inference and simultaneously improves solution quality compared to applying the same method to binarized versions of the problems. We believe the strong performance of our method on these non-binary benchmarks effectively demonstrates its capability and its advantages over prior neural solvers.
> >
> > [1] Zeng, H., Wang, J., Das, A., He, J., Han, K., Hu, H., & Sun, M. (2024, August). Effective generation of feasible solutions for integer programming via guided diffusion. In Proceedings of the 30th ACM SIGKDD Conference on Knowledge Discovery and Data Mining (pp. 4107-4118).

---

> > > ### Author Response · Authors · 2025-11-27
> > >
> > > As we enter the final week of the discussion period, we would like to once again convey our sincere gratitude for your thoughtful and constructive feedback. We also wish to gently remind you that we have submitted a detailed response to your comments. At your convenience, we would be deeply grateful if you could kindly let us know whether our clarifications have satisfactorily addressed your concerns. Please do not hesitate to share any further questions or suggestions—we greatly value your insight and engagement.

---

### Official Review · Reviewer_GgcN · 2025-10-31

**Soundness:** 3
**Presentation:** 1
**Contribution:** 3
**Rating:** 4
**Confidence:** 3

**Summary:**

This paper presents a method to design a ML-based heuristics for MILPs, cast as
drawing samples from a distribution over solution candidates. The underlying
probabilistic generative models used in this paper are variants of flows
(consistency models, short-cut models and mean-flow models) that are designed to
decode (relaxations of) solutions in very few denoising steps. They operate in
feature (dense) space (ie embedding space) as opposed to most ML methods fo COs
based on generative models which operate in solution (ie discrete) space. The
paper focus on MILPs with non-binary integer values.


In order to be able to work in high-dimensional dense spaces, this method
requires encoding and decoding program and candidate solutions.

Encoding is learned by a contrastive divergence model inspired by CLIP. This method is described
briefly and lacks explanation to properly understandhow it is actually performed.

For decoding, a simple classifier is trained to recover the discrete values
corresponding to the dense values predicted. This is complemented by IIP  in order to
encourage the model to predict real values as closed as possible to integer
values.


Models based on variants of flows (CMILP/SCMILP/MFILP) are presented in a very terse and lack definitions that can help understanding what they are about.
I am wondering what is the use if these paragraphs. Either they should only refer papers introducing flow variants, either they should be written in such a way as readers can understand (at least) the definition.
For instance, $\epsilon$ in CMILP is not defined, so the consistency constraint is simply impossible to understand. especially when models differ in time limits (for some t=0 is data for others it is noise).
The same can be said about SCMILP (eg /d/ is not defined) and MFILP ($\eta$ ?)
I suggest that only one variant be detailed in the main body of the article, and delegate the other two to the appendix.
This would leave room to understand the models and how they are adapted for the task at hand.

The MGD method is a nice addition the paper. I would recommend to use different letters than $\beta,\eta$ which are already used in the paper.



typos:
- l. 143  improper font for n/m
- l. 148 bt -> by
- l. 171 MES not defined
- l. 228  parentheses are imbalanced in Equation 4. Time scheduler is not defined.
- l.342 Is (Nair et al., 2021) the correct citation for PaS?

**Strengths:**

1. The paper presents a complete pipeline that can handle non-binary integer variables.
2. the global architecture relies on an encoder and a decoder to go from solution space to feature space. Once they are trained, the architecture is agnostic to the generative model (diffusion, consistency, short-cut, mean-flow)

**Weaknesses:**

- The contribution is essentially the pipeline encoder/generative mode/decoder but the paper has 3 sections (3.2,3.3,3.4) on fast flow models that are either very difficult to understand if the reader is not familiar with these formalism, or uninformative otherwise: notations are not introduced and there is no explanation.
- The training of the architecture is not performed end-to-end
- Result tables could be presented in a friendlier way, for instance by using bold face to help identifying the best systems.

**Questions:**

- Could training be performed end-to-end (encoder, generative model, decoder)?
- Could at least the encoder and the decoder be trained jointly (VAEs come to mind)?
- About IIP at inference time, how many times is *multiple* times (l. 206)?

---

> ### Author Response · Authors · 2025-11-22
>
> **Q1: The use of the introduction of fast diffusion solvers.**
>
> A1: Thank you very much for raising this concern. A key contribution of our work is the introduction of one-step diffusion models for solving non-binary integer linear programming problems. We believe it is essential to describe these solvers and their mechanisms to help readers understand the approach. At the same time, because our framework employs three diffusion models, the original description was necessarily brief due to page limits.
>
> To improve clarity, we have restructured the paper by moving the detailed descriptions of the shortcut model and mean flow model to the appendix, while providing a more thorough introduction of all diffusion solvers in the main text. We believe this revision significantly enhances the readability of our paper and allows readers to better understand the contributions. Additionally, we have addressed the other issues noted in the revision. We sincerely thank the reviewers again for these constructive suggestions.
>
>
> **Q2: Could training be performed end-to-end (encoder, generative model, decoder)? Could at least the encoder and the decoder be trained jointly (VAEs come to mind)?**
>
> A2: Thank you very much for raising these questions. Firstly, the simple answer is yes—our pipeline can in principle be trained end-to-end, and it can also be reformulated to incorporate a VAE-style architecture as you suggested. Secondly, we would love to state this once again that our decision to train the encoder separately stems from our intention to adopt a CLIP-style framework, which has been widely demonstrated to produce strong, well-aligned representations. Our goal is to learn an encoder that extracts meaningful and paired features from both MILP instances and their solutions, providing an effective foundation for the diffusion process. As you noted, a VAE represents an alternative paradigm for feature extraction, and both approaches have the potential to improve performance. Due to time limitation, we didn't run experiments with a VAE structure.
>
> To make our statements more convincing, we would love to show you some of the results by training end-to-end. From the experiment results, we can see that by training end-to-end, the model failed to capture effective features. Both the gap and feasibility are inferior compared to our original setting. We also observe that although the diffusion loss converges well under this scheme, both the reconstruction loss and the feasibility penalty remain high. This indicates that when trained jointly, the encoder tends to exploit shortcuts that minimize diffusion loss without learning meaningful structure—ultimately harming the solver’s overall performance. In contrast, pretraining the encoder ensures that the model learns robust, informative features before diffusion training begins. Based on these observations, we conclude that pretraining the encoder is essential for stable training and strong final performance.
>
> |              | Ours | End-to-end |
> | ------------ | ---- | ---------- |
> | CMILP(Ours)  |Gap 16.5% / Time 2.6s / S.Fea 69.2% / D.Fea 88.0%|Gap 54.8% / Time 2.5s / S.Feas 12.2% / D.Feas 18% |
> | SCMILP(Ours) |Gap 12.2% / Time 2.0s / S.Fea 42.4% / D.Fea 78.0%|Gap 57.2% / Time 2.9s / S.Feas 12.2% / D.Feas 11%|
> | MFILP(Ours)  |Gap 12.1% / Time 2.1s / S.Fea 70.5% / D.Fea 90.0%|Gap 46.5% / Time 2.5s / S.Feas 20.2% / D.Feas 26%|
>
> **Q3: About IIP at inference time, how many times is multiple times (l. 206)?**
>
> A3: Thank you very much for raising this question. In practice, we apply only 2–3 iterations of IIP during inference, and this is sufficient to bring the predictions very close to integer values. This behavior is also reflected in the illustration of the IIP layer in our paper.
>
> To further demonstrate the effect of the number of IIP iterations, we conducted an additional experiment on the IM-(50, 5, 2) dataset, using MFILP as a reference. The results show a consistent trend: increasing the number of IIP iterations leads to higher feasibility rates. This aligns with our analysis that appropriately applying IIP helps refine near-integer solutions and guides the model toward feasible regions more effectively.
>
> Overall, these findings confirm that controlled application of IIP meaningfully improves feasibility without compromising stability.
>
> |     | k=1 | k=2 | k=3 |
> | --- | --- | --- | --- |
> |MFILP|Gap 11.66% / S. Feas 68.1% / D. Feas 84%|Gap 12.02% / S. Feas 64.8% / D. Feas 87%|Gap 11.40% / S. Feas 72.9% / D. Feas 90%|

---

> > ### Author Response · Authors · 2025-11-27
> >
> > As we approach the final week of the discussion period, we would like to extend our sincere thanks once more for your thoughtful feedback. We would like to gently remind you that we have submitted a detailed response to your comments. When you have a moment, we would be grateful if you could let us know whether our explanations have resolved your concerns. Please feel free to share any additional questions or thoughts—we truly appreciate your engagement.

---

> > > ### Comment · Reviewer_GgcN · 2025-11-28
> > >
> > > Thank you for the detailed response (to my review and others').
> > >
> > > I think that the editorial work the author has made to improve readability will pay off for the next iteration.
> > >
> > > I understand that training end-to-end may be difficult, due to the shortcut effect leading the NN to memorize the training set.
> > > However, without additional information about the experimental setup of end-to-end training I cannot fully agree with the author's claim.
> > > Moreover, one could design a hybrid system: end-to-end training with CLIP embeddings used as initializations or stabilization points to prevent shortcut effects.

---

> > > > ### Author Response · Authors · 2025-11-30
> > > >
> > > > Thank you very much for your follow-up response and for recognizing the improvements we have made to the manuscript.
> > > >
> > > > Regarding your concern about our end-to-end training experiments, we would like to clarify that our end-to-end setup strictly follows the same experimental configuration as the CLIP-style training used in our method. The only modification is that the encoder parameters are made trainable. This design is intentional: it provides a clean and controlled comparison that isolates the effect of full end-to-end optimization. As shown in the results provided in our earlier response, directly training the entire model end-to-end consistently leads to a noticeable degradation in performance. This confirms that the CLIP-style separation is not an arbitrary choice but an essential part of achieving strong results.
> > > >
> > > > Regarding hybrid training, we additionally evaluated a variant where the encoder is initialized from a pretrained checkpoint before being fine-tuned. The results are shown as follows. Although this initialization provides some improvement over training from scratch, its performance remains substantially below that of the CLIP-type training. Taken together, these results demonstrate that (1) initialization helps but is insufficient, and (2) CLIP-style pretraining is a critical component that enables our latent-space formulation to function effectively.
> > > >
> > > > We sincerely appreciate the reviewer’s attention to this issue. Based on the comprehensive comparisons—including end-to-end training, hybrid initialization, and our CLIP-style approach—we believe we have thoroughly addressed the concern and provided clear empirical evidence supporting our design choice.
> > > >
> > > > We hope that these clarifications fully resolve your questions.
> > > > |              | Ours                                              | End-to-end                                        |hybrid|
> > > > | ------------ | ------------------------------------------------- | ------------------------------------------------- | --- |
> > > > | CMILP(Ours)  | Gap 16.5% / Time 2.6s / S.Fea 69.2% / D.Fea 88.0% | Gap 54.8% / Time 2.5s / S.Feas 12.2% / D.Feas 18% |Gap 48.3% / Time 2.2s / S.Feas 12.6% / D.Feas 19%|
> > > > | SCMILP(Ours) | Gap 12.2% / Time 2.0s / S.Fea 42.4% / D.Fea 78.0% | Gap 57.2% / Time 2.9s / S.Feas 12.2% / D.Feas 11% |Gap 45.7% / Time 2.3s / S.Feas 12.1% / D.Feas 19%|
> > > > | MFILP(Ours)  | Gap 12.1% / Time 2.1s / S.Fea 70.5% / D.Fea 90.0% | Gap 46.5% / Time 2.5s / S.Feas 20.2% / D.Feas 26% |Gap 54.9% / Time 2.3s / S.Feas 17.8% / D.Feas 32%|

---

### Official Review · Reviewer_3HsG · 2025-11-01

**Soundness:** 3
**Presentation:** 3
**Contribution:** 2
**Rating:** 6
**Confidence:** 4

**Summary:**

This paper proposes a novel class of one-step diffusion solvers for ILP, particularly for general integer variables. Building upon the observation that existing diffusion-based approaches (e.g., IP-guided DDPM/DDIM) can be viewed as gradient descent with only a single optimization step, the authors introduce three variants (CMILP, SCMILP, and MFILP). In addition, the paper proposes an objective-guided sampling strategy, integrating the optimization objective and feasibility penalties directly into the sampling stage via MGD. Extensive experiments on binary and general ILP benchmarks show that the proposed methods achieve higher sample feasibility and faster inference than both traditional solvers and IP-guided diffusion baselines, while maintaining acceptable optimality gaps.

**Strengths:**

The paper reinterprets diffusion-based optimization through a unified framework that connects consistency modeling, flow matching, and mean flow formulations within ILP. This conceptual reframing is novel, intellectually stimulating, and provides a coherent perspective on how generative diffusion dynamics can be viewed as gradient-based optimization.

Prior diffusion-based optimization methods were primarily restricted to binary MILPs. The paper’s treatment of general integer variables through the Iterative Integer Projection (IIP) mechanism and smooth surrogate gradients meaningfully broadens the applicability of diffusion models to a wider class of combinatorial problems.

Experimental results demonstrate consistent improvements in sample feasibility and substantial reductions in inference time compared to IP-guided diffusion baselines. The ability to produce feasible solutions in seconds rather than minutes constitutes a clear practical advantage, particularly for time-sensitive applications.

**Weaknesses:**

1. **Overstated novelty claim:** The claim of being the first to extend neural solver to the non-binary ILP case is overstated. Prior work [1] already introduced differentiable integer feasibility layers applicable to general MINLPs, including ILPs as a subset, and provided experiments on integer linear programs. The current contribution is better characterized as integrating such continuous-to-integer mappings within diffusion-based guidance.
2. **Limited optimality quality:** While the proposed methods achieve high feasibility rates, the optimality gaps remain significantly larger than those of traditional solvers such as Gurobi or SCIP. The models tend to produce feasible but suboptimal solutions, positioning them closer to heuristic generators rather than competitive optimization solvers.
3. **Unfair runtime comparison:** The runtime advantage reported against traditional solvers is partly a result of comparing approximate solutions to exact ones. A fairer comparison would fix a target optimality gap or time limit for all methods, as state-of-the-art solvers can also produce approximate feasible solutions within seconds under relaxed stopping criteria.
4. **Training data cost:** The paper mentions using approximately 500 optimal and near-optimal solutions per instance for training but does not discuss the computational cost of generating this data. If the time spent obtaining these labeled solutions is comparable to or exceeds that of running traditional solvers, the overall efficiency advantage of the proposed framework becomes questionable. A discussion of the data acquisition cost relative to inference savings would provide a more balanced assessment.
5. **Limited sensitivity analysis:**
While the experimental results are convincing, the paper would benefit from additional sensitivity studies on a few key hyperparameters, such as the number of IIP iterations or the momentum coefficient in MGD, to better understand their impact on feasibility and optimality. Nevertheless, the absence of such analyses does not substantially undermine the main empirical findings.

[1] Tang, B., Khalil, E. B., & Drgoňa, J. (2024). Learning to Optimize for Mixed-Integer Non-linear Programming with Feasibility Guarantees. arXiv preprint arXiv:2410.11061.

**Questions:**

1. The feasibility of the proposed methods appears highly dependent on the feasibility penalty term. Could the authors report sensitivity analyses over this coefficient (e.g., feasibility vs. penalty weight curves) to clarify how robust the model is to hyperparameter tuning?
2. The Iterative Integer Projection (IIP) function is differentiable but non-convex. Have the authors observed any convergence issues or gradient instability when applying IIP in practice?
3. The Iterative Integer Projection (IIP) layer is applied only once during training for efficiency, whereas multiple iterations are used during inference. This introduces a potential train–test discrepancy, since the training loss is computed in the continuous domain without enforcing full convergence to integers. Have the authors observed any degradation in performance or feasibility due to this mismatch?
4. For a fair runtime comparison, did the authors allow solvers like Gurobi or SCIP to terminate early under approximate optimality gaps (e.g., 1% or 5%)? If not, could the authors provide runtime comparisons under equivalent solution quality thresholds?
5. The paper claims that the proposed diffusion framework is more parameter-efficient than L2O-MIMLP. However, this comparison appears inconsistent: L2O-MIMLP [1] employs a simple MLP-based architecture, whereas the proposed model uses a Transformer backbone with contrastive pretraining and diffusion-based components, which likely involve significantly more parameters.

[1] Tang, B., Khalil, E. B., & Drgoňa, J. (2024). Learning to Optimize for Mixed-Integer Non-linear Programming with Feasibility Guarantees. arXiv preprint arXiv:2410.11061.

---

> ### Author Response · Authors · 2025-11-22
>
> **Q1: Could the authors report sensitivity analyses over this coefficient (e.g., feasibility vs. penalty weight curves) to clarify how robust the model is to hyperparameter tuning?**
>
> A1: Thank you for raising this question. We conducted an ablation study on the IM-(50, 5, 2) dataset to evaluate our model’s sensitivity to the penalty coefficient. The results show a clear trend: if the penalty coefficient is too small, the feasibility rate drops substantially; if it is too large, training becomes unstable and the model struggles to learn sufficient features for both diffusion and reconstruction. Importantly, within a broad intermediate range, the model remains robust and achieves consistently strong performance. While fine-tuning the coefficient can provide marginal gains, the improvement is not significant, suggesting that our method is stable and practical for real-world scenarios without extensive hyperparameter search.
>
> We set the penalty coefficient at the order of 1000 because only solutions with feasibility gaps lower than $10^{-5}$ as feasible solutions. To ensure that the feasibility penalty is on a comparable scale with the diffusion loss, a coefficient of this magnitude is necessary. Overall, as long as the coefficient remains within a reasonable scale, the feasibility penalty is effectively enforced.
>
> |        | penalty coef = 1000 | penalty coef = 2000 | penalty coef = 3000 | penalty coef = 5000|
> | ------ | ------------------- | ------------------- | -------------- | ---|
> | CMILP  |Gap 15.04% / Time 2.8s / S. Feas 51.0% / D. Feas 81.0% |Gap 15.0% / Time 2.6s / S. Feas 69.2% / D. Feas 89.0%|Gap 12.2%/ Time 2.8s / S. Feas 63.7% / D. Feas 88%|Gap 16.5% / Time 2.6s / S. Feas 65.7% / D. Feas 85%|
> | SCMILP |Gap 14.10% / Time 2.4s / S.Feas 26.8% / D. Feas 65%|Gap 12.2% / Time 2.0s / S. Feas 42.4% / D. Feas 78.0%| Gap 20.78% / Time 2.5s / S.Feas 33.1% / D. Feas 72% | Gap 25.78% / Time 2.5s / S.Feas 42.4% / D. Feas 78% |
> |MFILP|Gap 11.4% / Time 2.4s / S. Feas 72.9% / D. Feas 90%|Gap 12.1% / Time 2.1s / S. Feas 70.5% / D. Feas 90.0%|Gap 16.5% / Time 2.7s/S. Feas 50.4%/ D. Feas 85%|Gap 17.9% / Time 2.7s / S. Feas 68.3% / D. Feas 87%|
>
> **Q2: Have the authors observed any convergence issues or gradient instability when applying IIP in practice?**
>
> A2: Thank you for raising this concern. Firstly, in practice, we have not observed any gradient instability when applying the IIP layer. This is primarily because we apply only a single IIP iteration during training. While multiple iterations would indeed push the function closer to a hard rounding operator—potentially leading to very large gradients—one iteration keeps the gradient well-behaved.
>
> Specifically, the gradient function of the IIP layer is $1 - \cos(2\pi x)$, which lies in $[0, 2]$. The maximum gradient occurs in the middle between two subsequent integers, whereas near well-formed integer-like points, the gradient is small. As a result, the IIP layer doesn not introduce too much instability into training. Compared to the hard rounding function, whose gradient is undefined or infinite, our soft rounding function provides controlled and bounded gradients.
>
> Moreover, the IIP function can be viewed as a composition of sigmoid-like operations. Although sigmoid is concave in part of its domain, it is widely used in practice and is known to support stable optimization. Prior work [1] further analyzes a series of concave activations and shows that their use does not inherently harm convergence. Since we apply IIP only once and its gradients remain bounded, we do not observe exploding gradients or divergence issues in our experiments. Therefore, we believe that the IIP layer does not introduce overwhelming convergence or stability concerns.
>
> [1] Piotrowski, T. J., Cavalcante, R. L., & Gabor, M. (2024). Fixed points of nonnegative neural networks. Journal of Machine Learning Research, 25(139), 1-40.

---

> > ### Author Response · Authors · 2025-11-22
> >
> > **Q3: Have the authors observed any degradation in performance or feasibility due to the mismatch between the number of IIP iterasions in training and in testing?**
> >
> > A3: Thank you very much for raising this question. First, if the model’s prediction is already sufficiently close to an integer value—which is typically the case—the difference between the continuous prediction and its rounded counterpart is very small. In this regime, even a single IIP iteration effectively nudges the prediction toward the nearest integer without creating large penalties or instability. The purpose of the IIP layer is precisely to refine near-integer predictions, providing more accurate feasibility signals and improving the guidance given to the diffusion solver.
> >
> > Second, to further quantify the effect of the number of IIP iterations, we conducted an additional experiment on the IM-(50, 5, 2) dataset, using MFILP as a reference. The results show a clear trend: increasing the number of IIP iterations generally improves feasibility. This aligns with our earlier analysis that IIP improves the network’s ability to reach feasible regions. Importantly, the mismatch between using a single iteration during training and more iterations during testing does not degrade performance; instead, it improves feasibility at test time while preserving training stability.
> >
> > Such train–test mismatches are common in practice, where lighter operations are used during training to maintain stability, while stronger or repeated operations are applied during inference to improve final solution quality. Therefore, we believe that our choice of using a single iteration during training and multiple iterations during testing is well justified and does not negatively affect overall performance.
> >
> > |     | k=1 | k=2 | k=3 |
> > | --- | --- | --- | --- |
> > |MFILP|Gap 11.66% / S. Feas 68.1% / D. Feas 84%|Gap 12.02% / S. Feas 64.8% / D. Feas 87%|Gap 11.40% / S. Feas 72.9% / D. Feas 90%|
> >
> > **Q4: For a fair runtime comparison, did the authors allow solvers like Gurobi or SCIP to terminate early under approximate optimality gaps (e.g., 1% or 5%)? If not, could the authors provide runtime comparisons under equivalent solution quality thresholds?**
> >
> > A4: Thank you very much for raising this concern. To provide a fair comparison, we report results for exact solvers using a time budget comparable to that of our model. In the original paper, the reported time corresponds to performing inference for 30 sample solutions in parallel. For fairness, we use the time required to generate a single sample from our model as the time limit for the exact solvers.
> >
> > The results are summarized below. Under the same time constraints, our model achieves comparable or better performance than the exact solvers across all datasets. Notably, our model significantly outperforms COPT in these settings. While exact solvers remain highly effective at producing feasible solutions, our results further highlight the strength of our model in handling non-binary integer linear programming problems efficiently. This demonstrates that our approach provides a strong balance between solution quality and computational efficiency, particularly for challenging non-binary instances.
> >
> > |         | IM-(200, 5, 2)                     | IM-(100, 10, 2)                    | Random-(1000, 20, 2)|
> > | ------ | -------------------------------- | ---------------------------------- | ---------------------------------- |
> > | Gurobi  | Gap 0.0%, Time 46.6s, D. Feas 100% | Gap 0.0%, Time 53.3s, D. Feas 100% | Gap 0.0%, Time 18.1s, D. Feas 100%|
> > | Gurobi with limit  | Gap 12.1%, Time 3.9s, D. Feas 95% | Gap 13.5%, Time 4.2s, D. Feas 95% | Gap 2.2%, Time 2.6s, D. Feas 100% |
> > | COPT with limit | Gap 99.0%, Time 4.0s, D. Feas 100% | Gap 100%, Time 4.7s, D. Feas 100% | Gap 1e22%, Time 2.9s, D. Feas 100% |
> > | CMILP (1 sample time) | Gap 10.8%, Time 3.9s, D. Feas 90% | Gap 18.0%, Time 4.6s, D. Feas 67% | Gap 0.5%, Time 2.8s, D. Feas 87% |
> > | SCMILP (1 sample time) |  Gap 15.8%, Time 5.4s, D. Feas 86% | Gap 17.5%, Time 4.2s, D. Feas 62% | Gap 0.0%, Time 3.2s, D. Feas 89%  |
> > | MFILP (1 sample time)  | Gap 9.2%, Time 4.7s, D. Feas 90%  | Gap 16.1%, Time 4.6s, D. Feas 69% | Gap 0.0%, Time 2.6s, D. Feas 85% |

---

> > > ### Author Response · Authors · 2025-11-22
> > >
> > > **Q5: The paper claims that the proposed diffusion framework is more parameter-efficient than L2O-MIMLP. However, this comparison appears inconsistent: L2O-MIMLP [2] employs a simple MLP-based architecture, whereas the proposed model uses a Transformer backbone with contrastive pretraining and diffusion-based components, which likely involve significantly more parameters.**
> > >
> > > A5: Thank you very much for raising this question. While [2] introduces an additional neural network specifically to learn how to obtain feasible solutions, our approach achieves feasibility without adding any extra learnable parameters. Instead, we rely on the combination of our solver, carefully designed penalty terms, and guidance from the diffusion process to produce feasible solutions. This is precisely why we highlight that our model is more parameter-efficient. Importantly, both methods could use the same backbone architecture for the solver, so the distinction lies not in the backbone design but in how feasibility is enforced. By avoiding additional parameters, our model maintains simplicity and efficiency while still achieving competitive feasibility performance.
> > >
> > > [2] Tang, B., Khalil, E. B., & Drgoňa, J. (2024). Learning to Optimize for Mixed-Integer Non-linear Programming with Feasibility Guarantees. arXiv preprint arXiv:2410.11061.

---

> > > > ### Author Response · Authors · 2025-11-27
> > > >
> > > > As we enter the final week of the discussion period, we would like to once again express our sincere gratitude for your valuable feedback. We would also like to kindly remind you that we have posted a response to your comments. If convenient, we would greatly appreciate it if you could confirm whether our clarifications have addressed your concerns. We warmly welcome any further questions or discussion.

---

### Official Review · Reviewer_Eyya · 2025-11-08

**Soundness:** 2
**Presentation:** 2
**Contribution:** 3
**Rating:** 6
**Confidence:** 2

**Summary:**

The paper focuses on solving integer linear programming (ILP) problems with neural networks. To address long inference times and the difficulty of extending to non-binary integer problems, the authors propose three one-step diffusion-based ILP solvers, which are called CMILP, SCMILP, and MFILP, and propose to handle non-binary cases using an iterative integer projection (IIP) layer. To improve the sampling process, they further introduce a gradient-descent-based sampling method and a momentum mechanism into the objective-guided sampling of diffusion models.

**Strengths:**

1. Extending neural solvers from binary to non-binary ILPs is novel and useful.
2. The experimental comparisons—across different types of solvers, multiple metrics, and both binary and non-binary ILPs—are sufficient.

**Weaknesses:**

1. There are some typos—for example, “n variables and m constraints” in line 143.
2. Some descriptions could be clearer. For instance, not all symbols in Figure 1 are defined, which reduces readability. Also, the colors in Figure 3 should be made darker for better visibility.

**Questions:**

1. Why is the coefficient $\lambda$ applied only to the feasibility penalty term in Equation (2)? Why aren’t coefficients $λ_{recon}$ and $λ_{XXILP}$ used for the reconstruction error and the diffusion loss?
2. Can the method be applied to mixed ILPs? Are there any challenges?

---

> ### Author Response · Authors · 2025-11-22
>
> **Q1: There are some typos in the paper.**
>
> A1: Thank you very much for these helpful suggestions! We have corrected the equations and typos on line 143.
>
> Regarding clarification of Fig 1, we generally follow the symbols as defined in the Method section. This figure is intended to illustrate the overall pipeline of our model, which composes of (1) an encoder that takes a weighted bipartite graph derived from ILP instances as input, (2) various types of diffusion solvers operating in the latent space to solve the problem and (3) a solution decoder which maps the solutions back to the original space. In addition, we have incorporated an IIP layer to convert continuous outputs into integer solutions.
>
> We have also revised Figure 3 for better readability. Thank you once again for your valuable feedback.
>
> **Q2: Why is the coefficient lambda applied only to the feasibility penalty term in Equation (2)? Why aren’t coefficients lambda recon and lambdaXXILP used for the reconstruction error and the diffusion loss?**
>
> A2: Thank you very much for raising this question. We only introduced $\lambda_{penalty}$ because we want to keep the number of hyperparameters small to avoid extensive tuning. While it is possible to add additional hyperparameters to further optimize performance, our current results suggest that the model already performs satisfactorily. In general, the penalty coefficient is the most critical hyperparameter, as it determines how strongly we enforce feasibility. Moreover, because the feasibility term is tied to the ILP constraints, its scale naturally differs from the reconstruction and diffusion losses. For this reason, we consider the introduction of $\lambda_{\text{penalty}}$ both necessary and sufficient.
>
> We additionally conducted an ablation study on the penalty coefficient using the IM-(50, 5, 2) dataset. This is on the dataset IM-(50, 5, 2). The results show that performance remains relatively stable across a wide range of penalty values as long as the coefficient scale remains unchanged. If the penalty coefficient is set to 0, the training will collapse. The penalty coefficient is large because driving the feasibility gap to strict zero requires strong penalization. A large coefficient is also needed to bring the feasibility term to a scale comparable with the reconstruction and diffusion losses.
>
> |        | penalty coef = 1000 | penalty coef = 2000 | penalty coef = 3000 | penalty coef = 5000|
> | ------ | ------------------- | ------------------- | -------------- | ---|
> | CMILP  |Gap 15.04% / Time 2.8s / S. Feas 51.0% / D. Feas 81.0% |Gap 15.0% / Time 2.6s / S. Feas 69.2% / D. Feas 89.0%|Gap 12.2%/ Time 2.8s / S. Feas 63.7% / D. Feas 88%|Gap 16.5% / Time 2.6s / S. Feas 65.7% / D. Feas 85%|
> | SCMILP |Gap 14.10% / Time 2.4s / S.Feas 26.8% / D. Feas 65%|Gap 12.2% / Time 2.0s / S. Feas 42.4% / D. Feas 78.0%| Gap 20.78% / Time 2.5s / S.Feas 33.1% / D. Feas 72% | Gap 25.78% / Time 2.5s / S.Feas 42.4% / D. Feas 78% |
> |MFILP|Gap 11.4% / Time 2.4s / S. Feas 72.9% / D. Feas 90%|Gap 12.1% / Time 2.1s / S. Feas 70.5% / D. Feas 90.0%|Gap 16.5% / Time 2.7s/S. Feas 50.4%/ D. Feas 85%|Gap 17.9% / Time 2.7s / S. Feas 68.3% / D. Feas 87%|

---

> > ### Author Response · Authors · 2025-11-22
> >
> > **Q3: Can the method be applied to mixed ILPs? Are there any challenges?**
> >
> > A3: Thank you for much for raising this question. Yes our method can be extended to mixed ILP with a simple modification in the network architecture. In the original ILP setting, the encoder uses embeddings to encode discrete variables. For continuous variables, we introduce additional fully connected layers to perform the encoding. During decoding, the IIP layer is applied only to integer variables. Following FMIP, we adopt a weighted tri-partite graph structure to better capture the distinct characteristics of integer and continuous variables.
> >
> > Specifically, we choose item placement as a MILP dataset[1]. Due to the complexity of the problem, we need to resort to post-processing techniques to enforce feasibility. Our experiments show that even using FMIP’s publicly released checkpoints, no feasible solutions are produced without post-processing. Therefore, comparing all models under identical post-processing procedures is fair. Although our model performs slightly worse than FMIP in this setting, it is important to note that our architecture is significantly smaller. FMIP employs a 12-layer tri-partite GCN to model the diffusion process, while our approach uses the tri-partite GCN only in the encoder and relies on a transformer-based model for the diffusion process. Transformers are substantially more efficient than deep GCNs, and our method requires only a single inference step. We also report inference time excluding post-processing; this highlights that our one-step model is considerably more efficient.
> >
> > Finally, we would love to state that our method's major innovation is in its ability to deal with non-binary problems without resorting to binarification. The item placement dataset here is still a binary one regarding integer variables. Currently, there are limited open-source data generators for MILP with genuinely non-binary integer variables. We will consider its application to more complicated non-binary MILP in the future.
> >
> > |                | Pred&Search (600s)     | Apollo (800s)          | Neural Diving (400s)   |
> > | -------------- | ---------------------- | ---------------------- | ---------------------- |
> > | SL             | Gap -13.48%, Obj 15.34 | Gap -21.13%, Obj 14.16 | Gap -17.54%, Obj 14.62 |
> > | FMIP (Time 20.12s per instance)          | Gap -21.48%, Obj 13.92 | Gap -22.50%, Obj 13.74 | Gap -21.09%, Obj 13.99 |
> > | IP-Guided-Diff | Gap -14.21%, Obj 15.21 | Gap -19.17%, Obj 14.33 | Gap -17.48%, Obj 14.63 |
> > |MFILP (Ours, 0.62s per instance)|Gap -21.13%, Obj 14.16|Gap -21.48%, Obj 13.82|Gap -19.97%, Obj 14.33|
> >
> > [1] Gasse, M., Bowly, S., Cappart, Q., Charfreitag, J., Charlin, L., Chételat, D., ... & Kun, M. (2022, July). The machine learning for combinatorial optimization competition (ml4co): Results and insights. In NeurIPS 2021 competitions and demonstrations track (pp. 220-231). PMLR.

---

> > > ### Author Response · Authors · 2025-11-27
> > >
> > > As we enter the final week of the discussion period, we would like to sincerely thank you again for your valuable feedback. We would love to remind you that we have posted a response to your comments. We would appreciate it if you could kindly confirm whether our clarifications have addressed your concerns. We sincerely welcome any further questions or discussions.

---

### Meta-Review · Area_Chair_EVPE · 2026-01-08

**Summary:**

The paper proposes a class of one-step diffusion solvers (CMILP, SCMILP, MFILP) for general integer linear programming, utilizing a specialized iterative integer projection layer to handle non-binary variables. While the reviewers acknowledged the novelty of the framework and its speed advantages, several critical concerns remain unresolved. For example, the fairness of the comparison against traditional solvers remains contentious: the proposed method fails to demonstrate a clear advantage over Gurobi when strictly controlled for time or solution quality, as Gurobi often yields better feasibility under similar time constraints. Additionally, the computational cost of generating the large training dataset was not discussed. Consequently, the paper is not ready for publication at this time.

**Reviewer Concerns:**

### [Eyya]

All concerns are addressed.

### [3HsG]

The authors did not directly address the points raised in the "Weaknesses" section, though some were implicitly touched upon in their responses to the "Questions." A summary of the remaining issues follows:

W1: "Overstated novelty claim" is not addressed or argued.

W2 & W3 & Q4: Concerns regarding the solution quality and comparison fairness are not convincingly resolved. In the original paper, the comparison is hard to interpret as “superiority” because the axes being traded are different: classical solvers can guarantee feasibility/optimality (given enough time), while the proposed method is faster but does not guarantee either. The rebuttal adds a limited early-stop/time-budget experiment for Gurobi, but it still doesn’t establish a clear advantage: under comparable time budgets, Gurobi-with-limit appears at least competitive and often stronger on feasibility, while the proposed method may have comparable gap but lower feasibility. A convincing evaluation would include at least one of:
* Same time budget: proposed method achieves clearly better gap/feasibility;
* Same gap tolerance (with feasibility): proposed method is much faster;
* Solver integration (e.g., as a primal heuristic/callback): demonstrable reduction in end-to-end solve time.

None of these are demonstrated in a strong, systematic way in the paper or rebuttal.

W4: "Training data cost" is not addressed.

### [GgcN]

The reviewer’s main concern is the lack of end-to-end training (other issues are either minor or addressed). The authors demonstrated empirically that joint training degrades performance in their current setup, likely due to the complexity of the system and optimization challenges. The provided results are reasonable given the practical difficulties of training complex generative pipelines, but they do not definitively prove that a properly tuned end-to-end or VAE-style architecture is inherently inferior, suggesting this remains an open area for future architectural refinement rather than a settled design choice. Therefore, this concern is not convincingly addressed.

### [qDWz]

W3: Concerns regarding "large optimality gaps" are not fully addressed. See comments for 3HsG.

Other concerns are addressed.

**Reviewer Scores:**

Eyya: Likely to maintain the score of 6. While the authors effectively addressed the specific queries, the overall context of other reviews makes a score increase improbable.

3HsG: Expected to retain the score of 6 or potentially drop to 4, as the rebuttal failed to address critical concerns regarding novelty, comparison fairness, and data costs.

GgcN: Likely to remain at 4. While the authors provided empirical justification for their training approach, the reviewer's fundamental concern regarding the pipeline's architecture was not fully resolved.

qDWz: Likely to maintain the score of 4, as a significant concern remains only partially addressed.

---

### Decision · Program_Chairs · 2026-01-26

Reject